# Pre-post synaptic alignment through neuroligin-1 tunes synaptic transmission efficiency

Kalina T Haas[1,2†‡], Benjamin Compans[1,2†], Mathieu Letellier[1,2†], Thomas M Bartol[3], Dolors Grillo-Bosch[1,2], Terrence J Sejnowski[3], Matthieu Sainlos[1,2], Daniel Choquet[1,2,4], Olivier Thoumine[1,2], Eric Hosy[1,2*]

[1]Interdisciplinary Institute for Neuroscience, University of Bordeaux, UMR 5297, F-33000, Bordeaux, France; [2]Interdisciplinary Institute for Neuroscience, CNRS, UMR 5297, F-33000, Bordeaux, France; [3]Howard Hughes Medical Institute, Salk Institute for Biological Studies, La Jolla, United States; [4]Bordeaux Imaging Center, UMS 3420 CNRS, Université de Bordeaux, US4 INSERM, F-33000, Bordeaux, France

*For correspondence:
eric.hosy@u-bordeaux.fr

†These authors contributed
equally to this work

Present address: ‡Medical
Research Council Cancer Unit,
University of Cambridge,
Hutchison/MRC Research
Centre, Cambridge, United
Kingdom

Competing interests: The
authors declare that no
competing interests exist.

Reviewing editor: Gary L
Westbrook, Vollum Institute,
United States

**Abstract** The nanoscale organization of neurotransmitter receptors regarding pre-synaptic release sites is a fundamental determinant of the synaptic transmission amplitude and reliability. How modifications in the pre- and post-synaptic machinery alignments affects synaptic currents, has only been addressed with computer modelling. Using single molecule super-resolution microscopy, we found a strong spatial correlation between AMPA receptor (AMPAR) nanodomains and the post-synaptic adhesion protein neuroligin-1 (NLG1). Expression of a truncated form of NLG1 disrupted this correlation without affecting the intrinsic AMPAR organization, shifting the pre-synaptic release machinery away from AMPAR nanodomains. Electrophysiology in dissociated and organotypic hippocampal rodent cultures shows these treatments significantly decrease AMPAR-mediated miniature and EPSC amplitudes. Computer modelling predicts that ~100 nm lateral shift between AMPAR nanoclusters and glutamate release sites induces a significant reduction in AMPAR-mediated currents. Thus, our results suggest the synapses necessity to release glutamate precisely in front of AMPAR nanodomains, to maintain a high synaptic responses efficiency.

## Introduction

AMPA-type glutamate receptors (AMPARs) mediate the vast majority of fast excitatory synaptic transmission in the mammalian brain. AMPARs are stabilized at the post-synaptic density (PSD) by interactions with PDZ domain containing proteins such as PSD-95. AMPARs were initially thought to be homogeneously distributed throughout the PSD, but recent work based on super-resolution optical imaging (SRI) and electron microscopy has demonstrated that AMPARs are concentrated in small nanodomains around 80 nm in size, and containing 20 receptors on average (*Fukata et al., 2013*; *MacGillavry et al., 2013*; *Masugi-Tokita et al., 2007*; *Nair et al., 2013*). This specific mode of organization might be critical for synaptic transmission, depending on the relative positioning of pre-synaptic release sites with respect to AMPAR nanodomains. Indeed, previous studies showed that the glutamate content of a single presynaptic vesicle is not sufficient to activate the entire pool of AMPARs inside the PSD (*Liu et al., 1999*; *Raghavachari and Lisman, 2004*), suggesting that synaptic currents might be stronger if AMPARs were concentrated in front of pre-synaptic release sites rather than dispersed throughout the PSD. Moreover, mathematical models predict that when AMPARs are clustered in front of glutamate release sites, both the amplitude and the reliability of synaptic responses are improved (*Franks et al., 2002*, *Franks et al., 2003*; *Rusakov, 2001*;

*Rusakov and Kullmann, 1998*; *Tarusawa et al., 2009*). In contrast, when AMPAR clusters are not aligned with release sites, synaptic currents are predicted to be weaker and highly variable (*Tarusawa et al., 2009*). Therefore, it is critical to understand the spatial relationship between glutamate release sites and AMPAR domains at the nanoscale level.

Dual-color SRI offers a new way to analyze the alignment of pre- and post-synaptic elements underlying the intrinsic function of the synapse. Several studies have examined the nanoscale organization of various pre-synaptic proteins, including calcium channels, syntaxin, and neurexin (*Chamma et al., 2016*; *Ribrault et al., 2011*; *Schneider et al., 2015*), but these proteins did not display a clustered organization resembling that of AMPARs (*Hosy et al., 2014*; *Nair et al., 2013*). A recent study indicated however that the pre-synaptic active zone protein RIM is concentrated in small domains (*Tang et al., 2016*). Furthermore, this study showed that active glutamate release sites are co-localized with RIM and aligned with AMPAR nanodomains. This trans-synaptic 'nanocolumn' organization is regulated by long-term synaptic plasticity, highlighting its importance for the control of synaptic transmission. However, the molecular mechanisms underlying this alignment are still unknown, and the sensitivity of synaptic currents to mis-alignment has not been experimentally investigated.

One way to test the importance of pre- to post-synaptic alignment would be to destabilize the trans-synaptic organization and study its effect on synaptic transmission. It has been abundantly described that the adhesion complex neurexin-neuroligin forms a trans-synaptic bridge (*Südhof, 2008*). Pre-synaptic neurexin is implicated in active zone formation (*Missler et al., 2003*), while post-synaptic neuroligin-1 (NLG1) recruits PSD-95, NMDA receptors and AMPARs to partly structure the PSD (*Budreck et al., 2013*; *Graf et al., 2004*; *Heine et al., 2008*; *Mondin et al., 2011*). In particular, a C-terminally truncated NLG1 mutant, unable to bind PDZ domain containing PSD proteins (NLG1-ΔCter), was previously shown to prevent PSD-95 recruitment at newly formed synapses, and reduce AMPAR-mediated synaptic transmission (*Chih et al., 2005*; *Mondin et al., 2011*; *Nam and Chen, 2005*). Here, using dual-color SRI, we report that expression of the NLG1-ΔCter mutant or incubation with cell-permeant NLG1 C-terminal peptides suppress the co-localization of NLG1 and AMPAR nanodomains without changing the overall AMPAR nanoscale organization. In parallel, those treatments induce a physical shift between pre-synaptic RIM clusters and post-synaptic AMPAR nanodomains, associated with a significant decrease in AMPAR-mediated miniature and evoked EPSC amplitudes. We thus propose that neurexin-neuroligin mediated pre-post synaptic alignment tightly regulates synaptic efficacy.

## Results

### Expression of NLG1-ΔCter does not affect AMPAR nanoscale organization

To understand the role of NLG1 adhesion in AMPAR nano-organization, we performed direct STochastic Optical Reconstruction Microscopy (d-STORM; *Heilemann et al., 2008*) experiments on primary hippocampal neurons expressing either full length NLG1 (NLG1), or a NLG1 mutant with a truncation in the last 72 amino acids of the C-terminal domain (NLG1-ΔCter), both constructs carrying an N-terminal HA tag. NLG1-ΔCter has an intact extracellular domain allowing normal contacts with pre-synaptic partners such as neurexins (*Chih et al., 2005*; *Mondin et al., 2011*), but is unable to interact with cytoplasmic proteins within the PSD, and thus should behave as a dominant-negative mutant that uncouples trans-synaptic adhesion from the PSD. Given that AMPAR nanodomains are tightly associated with PSD-95 sites (*MacGillavry et al., 2013*; *Nair et al., 2013*), our rationale was that by disconnecting NLG1 from PSD-95, we would perturb the nanoscale positioning of AMPARs.

As previously described (*Mondin et al., 2011*; *Nam and Chen, 2005*), expression of both truncated and full length NLG1 (*Figure 1A and B*) increases spine density, revealing the synaptogenic role of NLG1 (n = 10; 12; 9 respectively; ANOVA p=0.0004). We then examined the effect of NLG1-ΔCter expression on AMPAR nano-organization. Surface AMPARs were detected by labeling live neurons with a primary antibody specific for the N-terminal domain of the AMPAR GluA2 subunit, followed by fixation and incubation with a secondary antibody conjugated to Alexa 647. Nanodomains are determined as clusters of AMPARs present inside the synapse and containing at least 5 receptors, based on single particle emission properties (see *Nair et al., 2013*). In control neurons

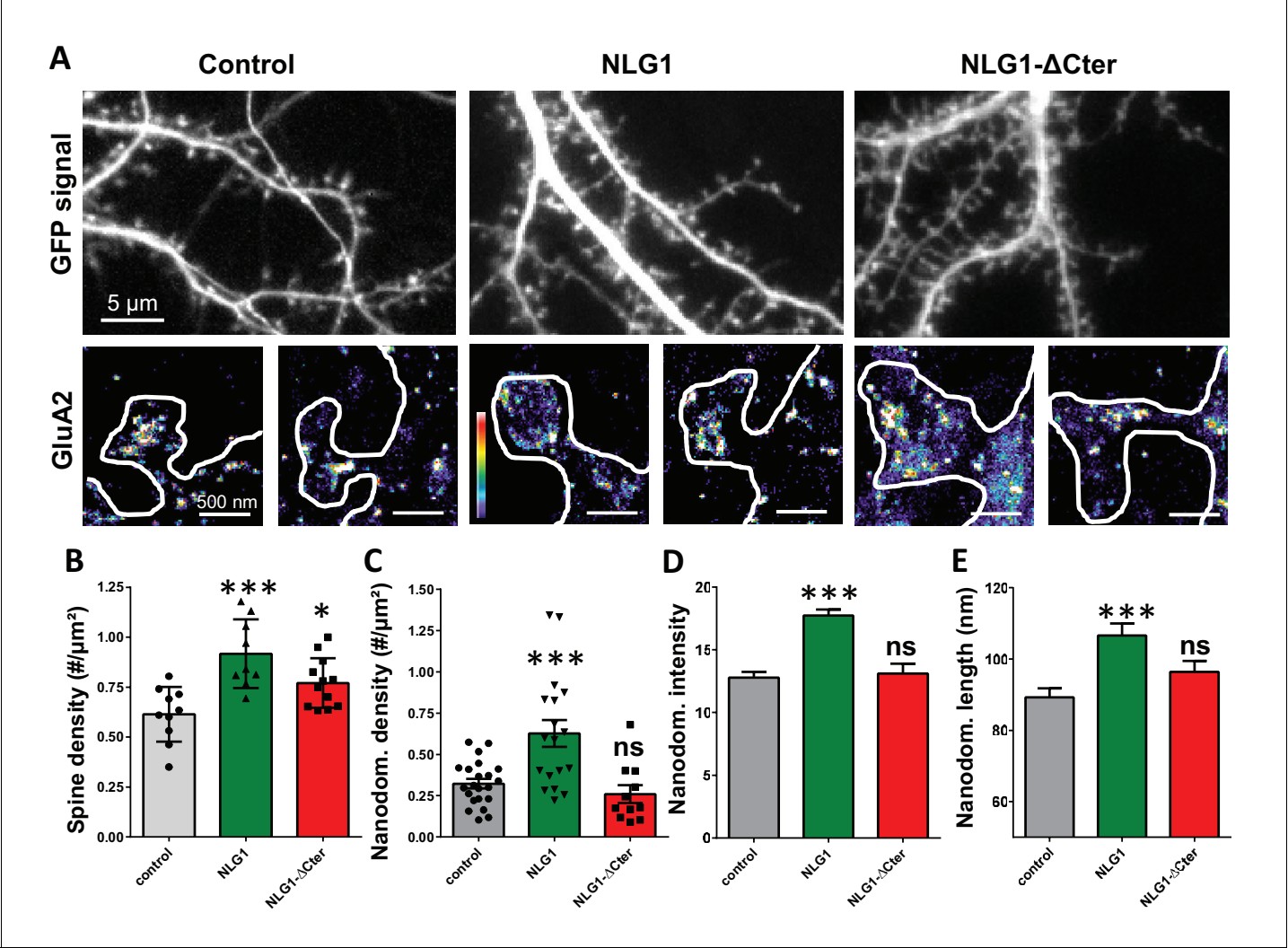

**Figure 1.** Expression of WT neuroligin but not NLG1-ΔCter affects AMPAR synaptic nano-organization. (**A**) Example of neurons transfected either with GFP, NLG1 + GFP or NLG1-ΔCter + GFP (from the left to the right), and two examples of AMPAR organization visualized with d-STORM technique. Intensity is color coded, scale go from 1 (purple) to 100 (white) detection per pixel. Average of spine density (**B**), AMPAR nanodomain density (**C**), nanodomains intensity expressed as number of receptors per nanodomain (**D**) and nanodomain length (**E**), on neuron expressing GFP, GFP +NLG1 and NLG1-ΔCter + GFP (n = 10; 9; 12 cells respectively; and between 200 to 500 individual domains).

The online version of this article includes the following figure supplement(s) for figure 1:

**Figure supplement 1.** Tessellation-based analysis reveals that synaptic AMPA receptor nanoscale organization is not affected by the expression of NLG1-ΔCter.

**Figure supplement 2.** NLG1-ΔCter expression does not affect AMPAR lateral mobility.

expressing GFP alone, we typically detected between 1 to 2 AMPAR nanodomains per synapse, with an average size of 90 ± 3 nm (*Figure 1D and E*), as reported previously (*Nair et al., 2013*). NLG1 overexpression increased the surface density of AMPAR nanodomains (*Figure 1C*), in agreement with the previous finding that NLG1 potentiates the formation of excitatory synapses (*Chih et al., 2005*; *Ko et al., 2009*; *Levinson et al., 2005*; *Mondin et al., 2011*). Furthermore, NLG1 overexpression led to a re-organization of AMPAR nanodomains by increasing both their size and AMPAR content (*Figure 1D and E*). Surprisingly, expression of NLG1-ΔCter for 3 days did not affect AMPAR nano-organization compared to the GFP control (*Figure 1C–E*, ANOVA post-test: p=0.72; 0.82 and 0.33 for figure C; D and E respectively). To validate this observation, we analyzed d-STORM images of AMPARs acquired on neurons expressing either GFP alone or GFP + NLG1-ΔCter, using a cluster quantification method based on Tessellation (*Levet et al., 2015*) (*Figure 1—figure*

*supplement 1A*). Through this analysis, we obtained an estimate of the number of endogenous GluA2-containing AMPARs per spine (*Figure 1—figure supplement 1B*), per nanodomain (*Figure 1—figure supplement 1C*) and the size of the nanodomains (*Figure 1—figure supplement 1D*). This analysis confirmed that NLG1-ΔCter expression does not affect the total amount of AMPARs per synapse, nor their organization in nanodomains.

Next, we assessed whether NLG1-ΔCter affected AMPAR surface diffusion, by measuring the lateral mobility of endogenous GluA2-containing AMPARs at the dendritic surface in live neurons using the universal Point Accumulation in Nanoscale Topography (u-PAINT) technique (*Giannone et al., 2010*; *Nair et al., 2013*) (*Figure 1—figure supplement 2*). We previously showed that AMPARs exhibit two diffusion profiles: the immobile fraction represents mostly AMPARs trapped by PSD scaffolding proteins in nanodomains, whereas the mobile fraction represents individual freely moving AMPARs (*Nair et al., 2013*). This lateral mobility is dependent on AMPAR complex composition, phosphorylation status and desensitization properties (*Compans et al., 2016*; *Constals et al., 2015*; *Hafner et al., 2015*; *Tomita et al., 2007*). Both the distribution of diffusion coefficients and the mobile fraction (i.e. proportion of AMPARs with diffusion coefficients above 0.01 µm²/s) were not significantly affected by NLG1-ΔCter, as compared to GFP-expressing neurons (*Figure 1—figure supplement 2*, n = 16 Ctrl and 21 NLG1-ΔCter, 2 sample t-test p=0.36), in agreement with previous findings using Quantum dots (*Mondin et al., 2011*). Thus, NLG1-ΔCter does not affect the equilibrium between diffusive and trapped AMPARs.

## Full-length NLG1 tightly co-localizes with AMPAR nanodomains

To examine the co-organization of AMPARs and NLG1 at the nanoscale level, we performed dual-colour SRI experiments. Endogenous GluA2 were live-labelled with a primary mouse monoclonal antibody and NLG1 or NLG1-ΔCter were live-labelled with a primary monoclonal rat anti-HA antibody. Neurons were then fixed, incubated with a secondary anti-mouse antibody conjugated to Alexa 532 and an anti-rat antibody coupled to Alexa 647, and processed for d-STORM (*Figure 2A*). Both AMPARs and NLG1 were detected as small synaptic and extra-synaptic clusters (*Figure 2B*). Qualitatively, we observed that the majority of synaptic NLG1 spots (green) overlapped with AMPAR nanodomains (purple). To precisely quantify the degree of co-localization between AMPAR and NLG1 clusters, we developed a method based on Manders' coefficients and bivariate nearest neighbor distance (see Materials and methods). Manders' coefficients have been widely used in diffraction-limited microscopy to quantify the co-localization between pairs of objects characterized by different fluorescent markers (*Manders et al., 1993*).

We first validated both the labelling efficiency and the co-localization by studying co-organization between HA-GluA1 (labelled with the same primary monoclonal rat anti-HA antibody, revealed with Alexa 647) and endogenous GluA2 (labelled with primary mouse monoclonal anti-GluA2 antibody, revealed with Alexa 532; *Figure 2—figure supplement 1*). These experiments have the advantage of using the same combination of antibodies as in *Figure 2*. Since AMPARs form heterotetramers in neurons, the labelling for GluA1 and GluA2 is expected to exhibit a high level of co-localization (*Figure 2—figure supplement 1*). The comparison between the size of single fluorescence emitters and both Alexa 532 and Alexa 647 labelled objects clearly demonstrates the presence of GluA1 and GluA2 clusters (*Figure 2—figure supplement 1B*). The distribution of bivariate nearest neighbor distances shows that 80% of HA-labelled GluA1 clusters have a GluA2 label within 80 nm (*Figure 2—figure supplement 1C*). The comparison of the experimental distribution of bivariate nearest neighbor distances to an in silico distribution obtained by randomization of cluster localization, clearly demonstrates that HA-GluA1 and GluA2 clusters display a high degree of co-localization (*Figure 2—figure supplement 1C*). Finally, the Manders' representation in (*Figure 2—figure supplement 1D*) reveals two points: first, only 5% of GluA1 and GluA2 object pairs have a null Manders' coefficient, reflecting that almost all Alexa 532-GluA2 labelled objects co-localize at least partly with Alexa 647-GluA1 labelled objects. Second, even though we co-labelled proteins belonging to the same cluster, less than 20% of objects exhibit a Manders' coefficient higher than 0.8, but 60% overlapped on an area larger than 50%. This likely originates from the fact that high levels of co-localization are difficult to reach in 2-color super-resolved images due to achromatism and the antibody size.

We then applied the Manders' analysis to examine the co-localization between AMPAR nanodomains and NLG1 clusters inside synapses (*Figure 2C,D and E*; green line). The similar area distribution between single fluorescence emitters and NLG1 clusters (*Figure 2C*) reveals that NLG1 does

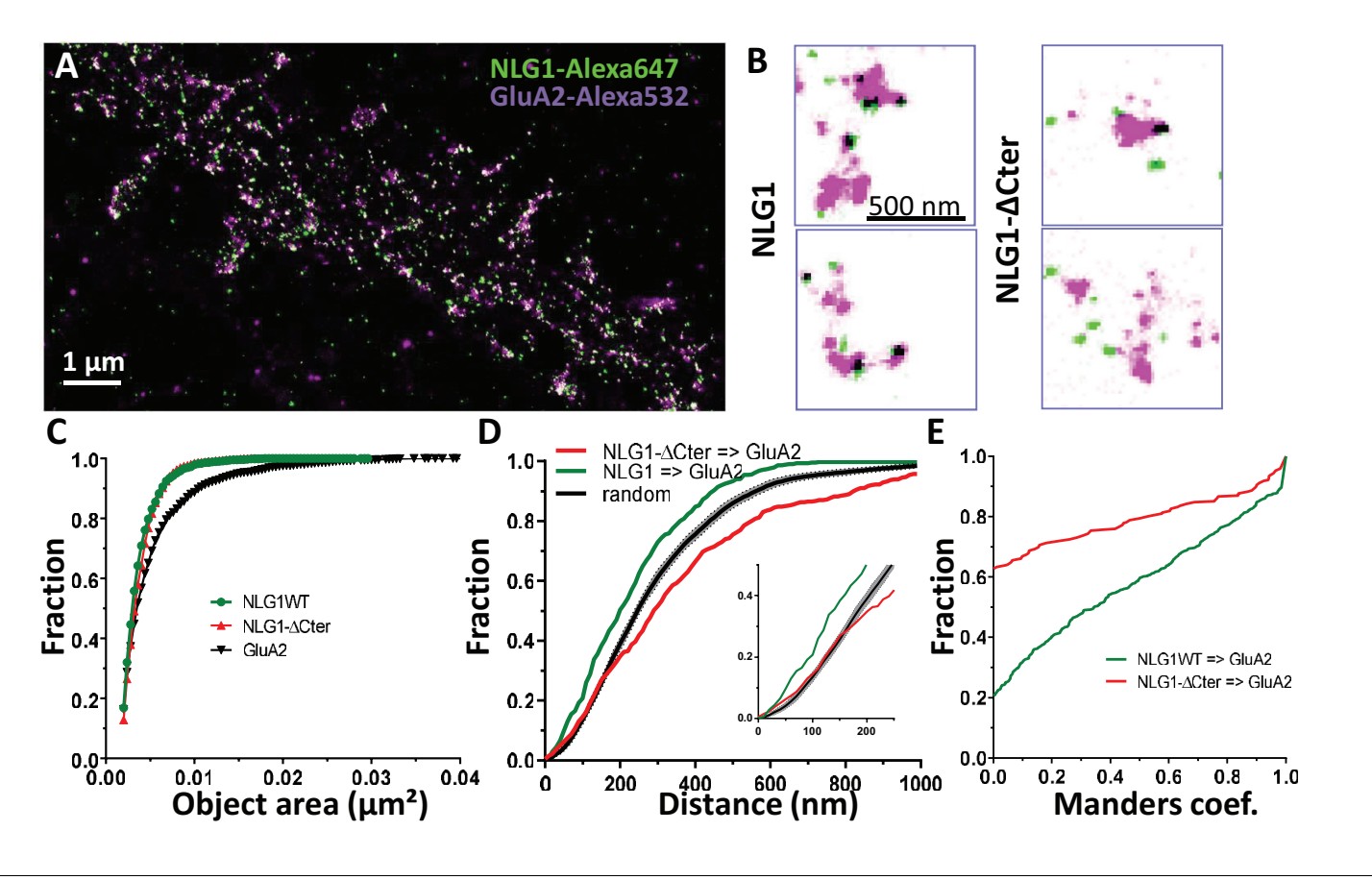

**Figure 2.** Delta C neuroligin does not co-localize with AMPAR nanoclusters. (**A**) Example of dual-color d-STORM super-resolution image of GluA2 containing AMPAR labelled with Alexa 532 nm and HA-tagged NLG1 labelled with Alexa 647 nm. (**B**) Examples of GluA2 and NLG1 (left) or GluA2 and NLG1-ΔCter (right) co-labeling of two synapses. Dark spots on the overlay image represent co-localizing pixels; NLG1 (in green) strongly co-localizes with AMPAR nanodomains (in purple). NLG1-ΔCter (in green) does not co-localize with AMPAR nanodomains (in purple). (**C, D** and **E**) presents the quantification of this co-localization. (**C**) The size distribution of NLG1 (green), NLG1-ΔCter (red) and GluA2 (black) super-resolved objects. The expression of NLG1-ΔCter does not affect the size of neuroligin 1 and GluA2 nanodomain objects. (**D**) Cumulative distribution of the measured (red) and randomized (dark) bivariate nearest neighbor distance between large object of GluA2 and NLG1-ΔCter. Green line represents the nearest neighbor distance between large object of GluA2 and neuroligin 1, demonstrating clustering as compared to random distribution. Insert represents a zoom on the 250 nm, approximate size of a PSD; NLG1-ΔCter and GluA2 nanodomain distance overlaps with the random distribution distance. (**E**) Manders' coefficients calculated between GluA2 nanodomains and NLG1-ΔCter (red) and between GluA2 nanodomains and NLG1 (green). More than 60% of AMPAR nanodomains are not co-localized with NLG1-ΔCter (n = 18 NLG1 and 12 NLG1-ΔCter cells respectively, corresponding to 516 and 312 independent pairs of objects).

The online version of this article includes the following figure supplement(s) for figure 2:

**Figure supplement 1.** Validation of the method to analyze super-resolved object co-localization with a dual labeling of AMPAR.

not form large domains inside synapses, but rather several small clusters, confirming previous findings with an alternative labelling strategy (*Chamma et al., 2016*). In contrast, the area distribution of AMPARs displays larger values, due to AMPAR clustering. For the remaining part of the analysis, we took into account all NLG1 objects, only selected AMPAR objects larger than 0.005 µm² (red dashed line *Figure 2—figure supplement 1B*), a threshold allowing the suppression of 80% of single emitters, to focus analysis only on AMPAR clusters (value of 0.0022 ± 0.0027 µm² which correspond to mean ± 1.5*STD). The centroid to centroid bivariate nearest neighbor distance distribution between NLG1 and AMPAR nanodomains was significantly smaller than that expected from an independent distribution (*Figure 2D*; green line, n = 512 pairs of co-localized objects), indicating a functional proximity between NLG1 and GluA2 clusters at the nanoscale. The difference between the two curves was already apparent in the first 100 nm, revealing a tight association between NLG1 and

AMPAR nanodomains at a short length scale. Finally, Manders' coefficients were calculated between each pair of objects (*Figure 2E*; green line, n = 512 pairs of objects). Only 20% of NLG1 clusters did not co-localize, even partially, with a GluA2 nanodomain, while 25% of NLG1 clusters co-localized to more than 80% with AMPAR nanodomains. These results reveal a fairly tight nanoscale co-organization between AMPAR and NLG1 within the synapse.

## NLG1-ΔCter is delocalized from AMPAR nanodomains

Next, we analyzed the distribution of AMPAR nanodomains with respect to NLG1-ΔCter (*Figure 2D and E*; 316 pairs of objects). As NLG1, NLG1-ΔCter was organized in small clusters (*Figure 2C*, red line) with an average size of 0.0042 ± 0.0018 μm$^2$, whereas AMPARs were distributed as both small and large objects. The bivariate nearest neighbor distances between NLG1-ΔCter and AMPAR clusters were significantly larger than for NLG1 and AMPAR nanodomains. Centroid to centroid bivariate nearest neighbor distance between all NLG1-ΔCter object and GluA2 clusters overlapped with a random distribution, at least for the first 250 nm corresponding to the PSD size (*Figure 2F*, red line). This led us to the conclusion that there is no preferential association of NLG1-ΔCter clusters with AMPAR nanodomains. This was confirmed by looking at the distribution of Manders' coefficients (*Figure 2G*, red line). Only 38% of NLG1-ΔCter co-localized at least partially with AMPARs, compared with 80% for NLG1, revealing that the NLG1 C-terminal truncation significantly decreases the association between NLG1 and AMPAR nanodomains (comparison with NLG1: 2 sample t-test p<0.0001).

## NLG1-ΔCter expression shifts the alignment between presynaptic RIM and post-synaptic AMPAR nanodomains

We then analyzed the effect of NLG1-ΔCter expression on the alignment between pre-synaptic RIM and post-synaptic AMPAR clusters. These two proteins are components of the trans-synaptic 'nanocolumns', which organize pre-synaptic release sites in front of post-synaptic AMPAR clusters (*Tang et al., 2016*). Endogenous RIM and AMPARs were labelled, while post-synaptic neurons expressed GFP, NLG1 or NLG1-ΔCter (*Figure 3A,B and C*). The bivariate nearest neighbor distances between RIM and AMPAR clusters was not affected by NLG1 expression, but was significantly larger when NLG1-ΔCter was expressed (*Figure 3B*, Anova p<0.001). The distribution of Manders' coefficients confirmed that NLG1-ΔCter expression decreased the RIM-AMPAR apposition (*Figure 3C*, Anova p<0.001). These results demonstrate a physical misalignment between the presynaptic marker RIM and the postsynaptic AMPAR clusters upon NLG1-ΔCter expression, while NLG1 expression did not display a similar effect.

## NLG1 C-terminus truncation impairs synaptic transmission

To estimate the effect of a decorrelation between AMPAR nanoclusters and pre-synaptic release sites on synaptic currents, we first recorded AMPAR-mediated miniature EPSCs on dissociated hippocampal cultures by electrophysiology, upon expression of NLG1-ΔCter + GFP, full length NLG1 + GFP, or GFP alone as a control (*Figure 4A–C*). NLG1-ΔCter expression for 3 days reduced the amplitude of AMPAR mEPSCs by 24% as compared to GFP expressing controls (*Figure 4C*, average ± SEM: 11.63 ± 0.63 for GFP, 8.956 ± 0.40 for NLG1-ΔCter, 13.05 ± 0.92 for NLG1; ANOVA p=0.0002). In contrast, NLG1 expression did not affect the miniature EPSC amplitude, even if as expected by the higher synapse number observed in *Figure 1A*, there was a significant increase in miniature frequency (*Figure 4—figure supplement 1A*). These results suggest that quantal synaptic transmission is not altered by NLG1 overexpression per se, but only when the linkage between NLG1-based adhesion and AMPAR clusters is perturbed.

To confirm these conclusions in a model system having more preserved synaptic connectivity, we used mouse organotypic hippocampal slice cultures, in which single CA1 neurons were co-electroporated with both soluble GFP and either NLG1 or NLG1-ΔCter. One week later, evoked whole-cell currents were recorded from electroporated neurons and from neighboring non-electroporated counterparts, upon stimulation of Schaffer collaterals (*Figure 4D–I* and *Figure 4—figure supplement 2*). In the presence of 2 mM extracellular calcium, we observed a ~4 fold increase in AMPAR-mediated EPSC amplitude (456.1% ± 80.0) in neurons overexpressing NLG1 compared to paired non-electroporated controls, reflecting the well-known synaptogenic effect of NLG1 (*Kwon et al.,*

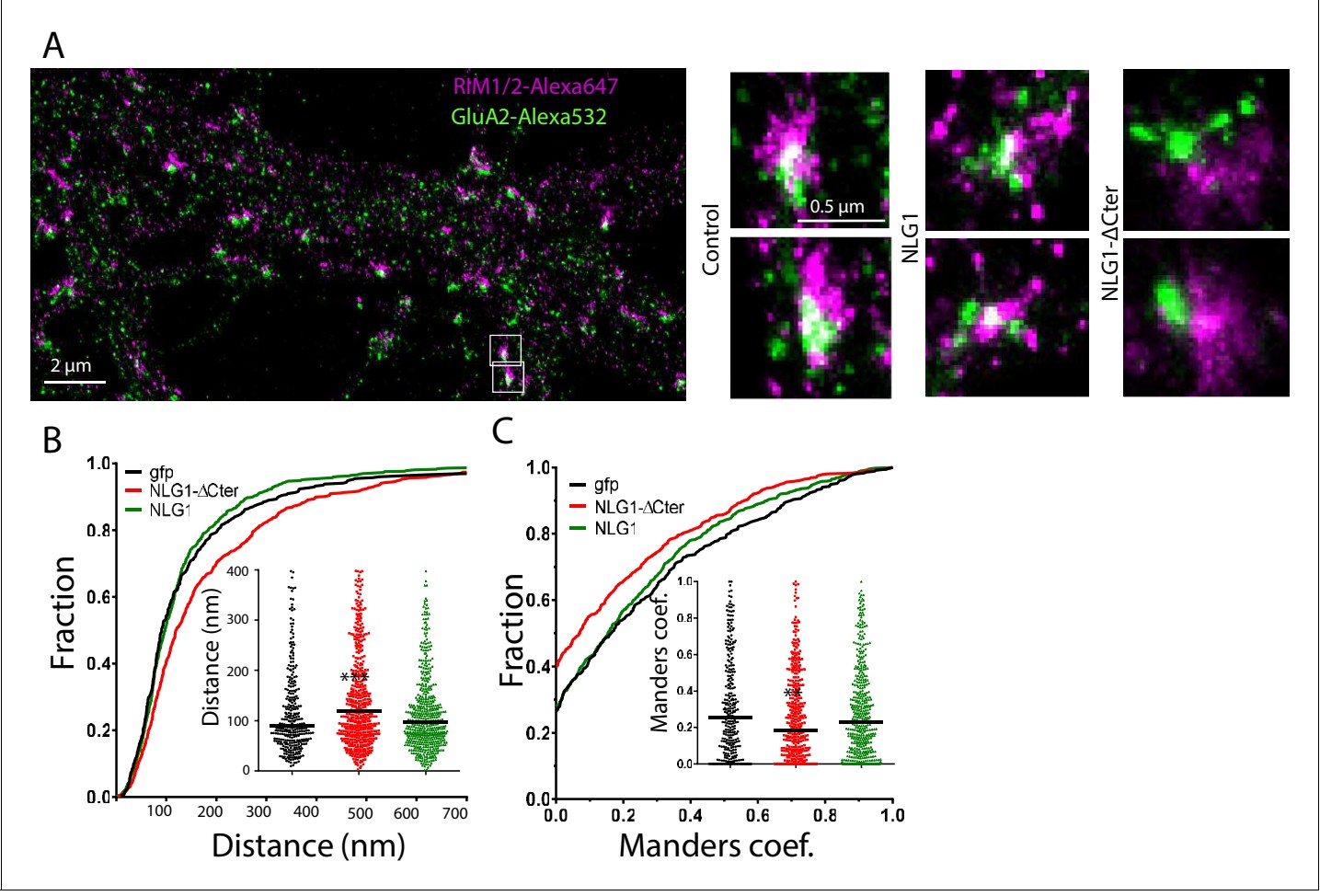

**Figure 3.** Delta C neuroligin expression decorrelates pre-synaptic RIM from AMPAR nanoclusters. (A) Example of dual-color d-STORM super-resolution image of GluA2 containing AMPAR labelled with Alexa 532 nm and RIM labelled with Alexa 647 nm. Right panels, examples of GluA2 and RIM co-labeling when post-synaptic neurons are transfected with GFP, NLG1 or NLG1-ΔCter respectively (from the left to the right). (B and C) presents the quantification of this co-localization. (B) Cumulative distribution of the bivariate nearest neighbor distance between GluA2 and RIM clusters when post-synaptic neuron expressed GFP (dark), NLG1 (green) or NLG1-ΔCter (red). These data demonstrate a loss of the pre-post synaptic alignment when NLG1-ΔCter is expressed. (C) Manders' coefficients calculated between GluA2 nanodomains and RIM clusters in function of the neuroligin wt or truncated form expression. NLG1-ΔCter expression significantly alters the co-localization (n = 9; 9 and 8 Control, NLG1-ΔCter and NLG1 cells respectively, corresponding to 354; 573 and 562 independent pairs of co-localization).

2012; *Shipman et al., 2011*). This effect was associated with a decrease in the coefficient of variation (CV) (*Figure 4—figure supplement 2*). This parameter is commonly related to both number of active synapses and their release probability (quantal content) but previous studies provided evidence for the contribution of post-synaptic parameters into the CV as for example the fraction of silent synapses (*Kerchner and Nicoll, 2008*; *Kullmann, 1994*) and ongoing fluctuations of AMPAR number at the synapse and their mobility (*Czöndör et al., 2012*; *Heine et al., 2008*; *Levy et al., 2015*). These results support the idea that NLG1 can both increase the number of functional synapses and stabilize AMPARs (*Levinson et al., 2005*; *Mondin et al., 2011*). In contrast, neurons expressing NLG1-ΔCter exhibited a limited ~2 fold increase in AMPAR-mediated EPSC amplitude (226.0 ± 59.1) and no change in CV compared to control neurons (*Figure 4—figure supplement 2*). Thus, when compared to NLG1, NLG1-ΔCter expressing neurons display lower EPSCs and higher CV, despite the fact that NLG1- ΔCter expression can still increase synapse number (*Figure 1B*). These joint effects can be explained by either a lack of recruitment of AMPARs at newly formed synapses (i.e. silent synapses) and/or a mis-alignment between pre- and post-synapses. Finally, there was no significant change in the paired-pulse ratio (PPR) upon expression of either NLG1 or NLG1-Δ

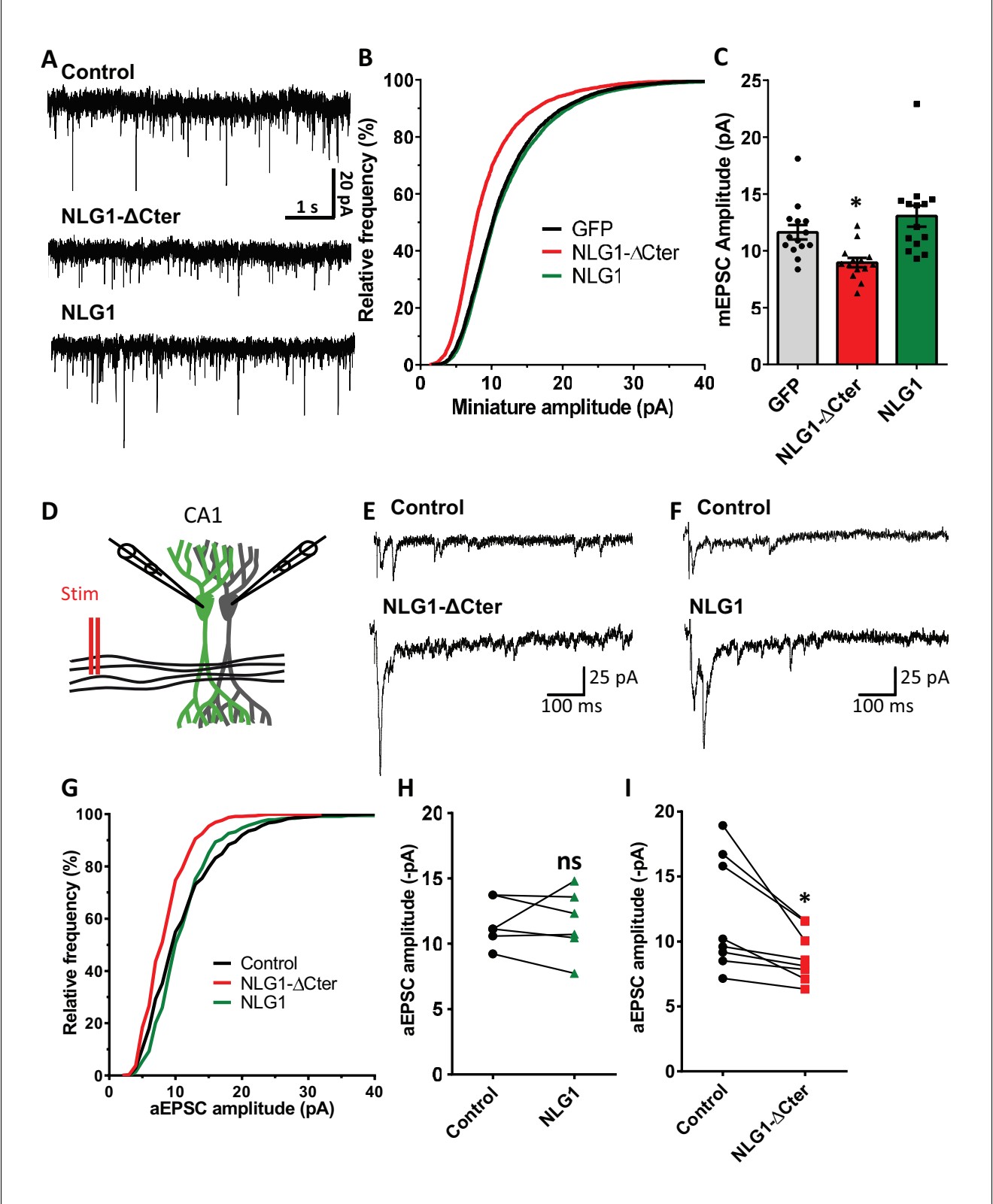

**Figure 4.** NLG1-ΔCter expression strongly impaired synaptic transmission efficacy. (**A**) Example of mEPSC traces recorded in cultured neurons expressing either GFP, GFP + NLG1-ΔCter or GFP + NLG1. (**B**) Cumulative distribution and (**C**) average of the mEPSC amplitude recorded on neurons expressing GFP (dark), GFP + NLG1-ΔCter (red) or GFP + NLG1 (green) (n = 14; 14 and 13 respectively). mEPSCs amplitude is decreased by 25% in neurons expressing NLG1-ΔCter. (**D**) Scheme of the patch clamp protocol used to record asynchronous EPSC on organotypic hippocampus slices. Two

*Figure 4 continued on next page*

*Figure 4 continued*

neighboring neurons are simultaneously patched, one transfected and one non transfected, Schaffer collateral connecting both neuron are then stimulated. (E and F) Representative traces of asynchronous EPSCs recorded in the presence of strontium, of either a control and a NLG1-ΔCter expressing neuron (E) or a control and a NLG1 (F). To avoid multi synaptic responses, 50 ms following the stimulation are excluded from the analysis. (G) Cumulative distribution of aEPSCs amplitude recorded from control (dark),or neurons expressing GFP + NLG1-ΔCter (red) or GFP + NLG1 (green) (n = 8 and 6 pairs of cells, respectively). Average of aEPSCs amplitude, with connection between the transfected cell and their respective neighboring non transfected control, when either GFP + NLG1 (H) or GFP + NLG1-ΔCter (I) are expressed. NLG1-ΔCter expression decreased by 25% the average aEPSCs amplitude.

The online version of this article includes the following figure supplement(s) for figure 4:

**Figure supplement 1.** Neuroligin 1 and NLG1-ΔCter expression have various effect on synaptic transmission properties both on neuronal cell culture and organotypic slices.

**Figure supplement 2.** EPSC amplitude from NLG1-ΔCter expressing neurons is decreased compared to neurons expressing NLG1.

**Figure supplement 3.** aEPSCs are decreased when NLG1-ΔCter is expressed in NLG1 KO background.

**Figure supplement 4.** Expression of NLG1-ΔCter does not affect paired-pulse ratio.

Cter (*Figure 4—figure supplement 4*; PPR for NLG1: control 1.63 ± 0.11, NLG1 1.36 ± 0.09, for NLG1-ΔCter), indicating no specific modification of the pre-synaptic release probability.

The amplitude of EPSCs recorded in calcium combines three parameters: the number of synapses (n), the release probability (p), and the quantal size (q), related to glutamate receptor structural organization, possibly including the alignment of AMPARs nanodomains with presynaptic release sites.. To test the idea that NLG1-ΔCter affects quantal size, extracellular calcium was replaced by strontium to trigger asynchronous pre-synaptic release following stimulation, thereby evoking a train of AMPAR-mediated EPSCs generated by glutamate release from single vesicles (*Goda and Stevens, 1994*). Both NLG1 and NLG1-ΔCter expression increased the frequency of asynchronous EPSCs compared to non-electroporated neurons, again reflecting the synaptogenic effect of NLG1 (*Figure 4—figure supplement 1B*). Moreover, as observed with miniatures in dissociated cultures, the amplitude of asynchronous EPSCs was reduced by 26% in neurons expressing NLG1-ΔCter but not in neurons expressing NLG1, when compared to control non-electroporated neurons (*Figure 4H and I*, for NLG1 expression: control: 11.59 ± 0.73, NLG1: 11.60 ± 1.03; for NLG1-ΔCter expression: control: 12.01 ± 1.56, NLG1-ΔCter: 8.893 ± 0.70; paired t-test p=0.99 and 0.018). Similar results were obtained on the NLG1 KO background (*Figure 4—figure supplement 3*), with a 32% decrease of asynchronous EPSC amplitude on NLG1-ΔCter expressing neurons relatively to unelectroporated counterparts. The similarity of the current decrease obtained in both the NLG1 WT and KO backgrounds, emphasizes the dominant negative role of NLG1-ΔCter which might compete with various NLG isoforms. Together, these results show that NLG1-ΔCter directly affects the quantal size, which is compatible with our concept of synaptic mis-alignment.

## Cell-permeant NLG1 biomimetic divalent ligand disrupts pre-post alignment and decrease AMPAR-mediated synaptic transmission

To confirm the role of NLG1 C-terminal interactions in aligning AMPARs in front of pre-synaptic release sites, without overexpressing NLG1-ΔCter mutants which might also affect pre-synaptic development, we used an alternative strategy. Based on our previous expertise to perturb interactions between stargazin and PSD-95 (*Sainlos et al., 2011*), we developed divalent biomimetic ligands comprising the 15 C-terminal amino acids of NLG1, conjugated to a TAT sequence to favor cell penetration. In contrast to the NLG1-ΔCter mutant, those ligands directly compete with endogenous neuroligins to bind PDZ domain containing scaffolding proteins at the post-synapse, without altering the binding of neuroligins to pre-synaptic proteins such as neurexins. Control non-sense ligands had the same structure but mutations in the sequence prevent interaction with PDZ domains. Incubation of 14 DIV hippocampal neurons for 1–2 days with NLG1 competing ligands decreased the co-localization between RIM and GluA2 containing AMPARs observed by dual-color d-STORM (*Figure 5A–C*), and reduced by 30% AMPAR-mediated mEPSC amplitudes (*Figure 5E*). In both assays, control non-sense TAT peptides had no effect compared to untreated neurons. Interestingly, NLG1 peptides did not alter AMPAR-mediated mEPSC frequency (*Figure 5F*), suggesting an exclusive post-synaptic effect. Overall, NLG1 competing peptides and NLG1-ΔCter expression had very

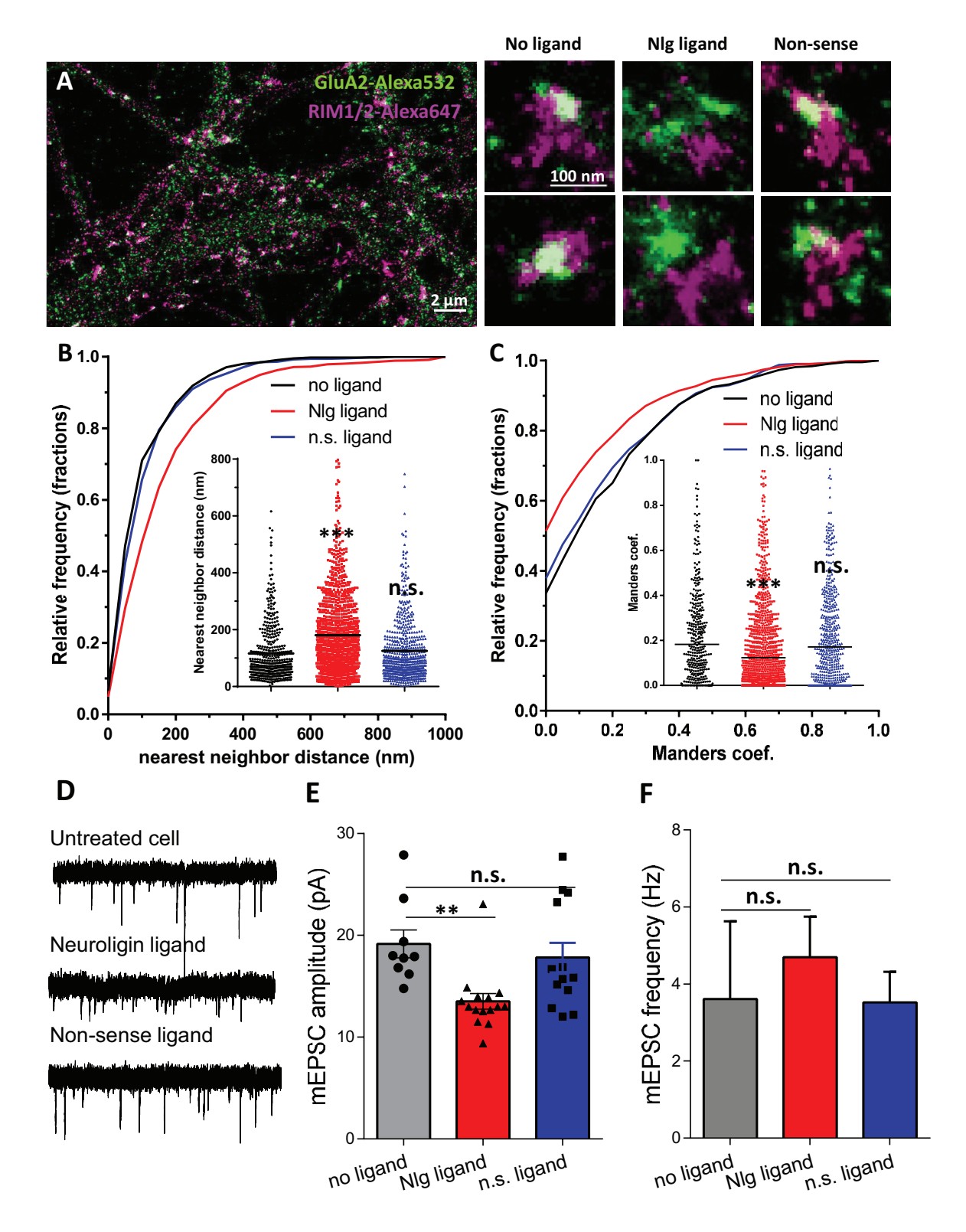

**Figure 5.** Acute disruption of PSD95-NLG interaction impaired both pre-post alignment and synaptic transmission. (**A**) Example of dual-color d-STORM image of GluA2 containing AMPAR labelled with Alexa 532 nm and RIM labelled with Alexa 647 nm. Right panels, examples of GluA2 and RIM co-labeling without ligand or after 1-day treatment with Nlg biomimetic ligand or non-sense ligand (from the left to the right). (**B** and **C**) presents the quantification of this co-localization. (**B**) Cumulative distribution of the bivariate nearest neighbor distance between GluA2 and RIM clusters without

*Figure 5 continued on next page*

*Figure 5 continued*

ligand (black), with NLG ligand NLG1 (green) or non-sense ligand (blue). These data demonstrate a loss of the pre-post synaptic alignment in the presence of NLG ligand. (C) Manders' coefficients calculated between GluA2 nanodomains and RIM clusters in function of ligand treatment (n = 4; 5 and 4 Control, NLG ligand and non-sense ligand respectively, corresponding to 451; 1311 and 640 independent pairs of co-localization). (D) Example of mEPSC traces recorded in cultured neurons without ligand, with NLG ligand or with non-sense ligand. (E and F) average of the mEPSC amplitude and amplitude recorded when neurons in culture are incubated without ligand (dark) or with either NLG ligand (red) or with non-sense ligand (blue) (n = 9; 15 and 13 respectively). mEPSCs amplitude is decreased by 30% in neurons incubated with NLG ligand.

similar effects on AMPAR positioning and synaptic responses, suggesting a common mechanism of action.

## Synaptic efficiency critically depends on the AMPAR nanodomains to glutamate release site distance

To examine theoretically the effects of delocalizing AMPAR nanodomains from pre-synaptic glutamate release sites, we performed Monte-Carlo-based simulations using the MCell software (*Figure 6*). The synaptic shape and perisynaptic environment were obtained from 3D electron microscopy images reconstructing the neuropil of a hippocampal CA1 stratum radiatum area, previously developed to model synaptic transmission (*Bartol et al., 2015a*, *Bartol et al., 2015b*; *Kinney et al., 2013*) (See Materials and methods). AMPAR chemical kinetic properties were obtained from a well-established model (*Jonas et al., 1993*) (*Figure 6B*) and the kinetic parameters were adjusted to fit both the recorded mEPSCs and the AMPAR organization map extracted from the d-STORM data (*Figure 1*, see Materials and methods [*Nair et al., 2013*]). In the simulations, the number of released glutamate molecules was fixed to 1500, 2000, 3000 or 4500, to be in the range of the estimated amount per pre-synaptic vesicle (*Savtchenko et al., 2013*). Simulations computed the number of open AMPARs, when vesicles containing the various glutamate quantities were released in front of a single AMPAR cluster, or up to 200 nm away from the cluster center, varied with a 50 nm increment (*Figure 6B*). As expected, simulated curves demonstrated that the glutamate content per pre-synaptic vesicle was positively correlated with the synaptic response (*Figure 6C*). Strikingly, the simulation further showed that the number of opened AMPAR was inversely correlated to the distance between the release site and AMPAR nanodomains (*Figure 6D*). A release of 3000 glutamate molecules, which is in the upper range of glutamate content per vesicle, at 150 nm from an AMPAR nanodomain, led to a decrease of almost 40% of the synaptic response. In this model, the release distance is measured from the center of the nanodomain, which has a 90 nm diameter size. Even at 50 nm from the centroid (i.e. at the close periphery of the nanodomain), a significant decrease of current amplitude was already observed for low glutamate content (*Figure 6D*). The expected decrease in synaptic current amplitude was also sensitive to the glutamate content (*Figure 6E*). Specifically, the 26% decrease of AMPAR-mediated current observed experimentally corresponds to a glutamate content of around 2000 molecules when the release site is localized 90 nm away from the nanodomain centroid (*Figure 6E*). Finally, in order to investigate the possible effect of a complex extracellular matrix and glial processes around the synapse, which might also regulate synaptic transmission (*Dityatev and Rusakov, 2011*), we varied the glutamate diffusion coefficient (*Figure 6—figure supplement 1*) or the density of astrocytic glutamate transporters (*Figure 6—figure supplement 2*). The first parameter drastically changed the synaptic transmission efficiency (i.e. the lower the glutamate diffusion coefficient, the higher AMPA currents) without altering the tight correlation between synaptic transmission and pre-post alignment. The second parameter revealed the independence of AMPAR-mediated currents with respect to the astrocyte-dependent clearance of the glutamate released from a single vesicle.

## Discussion

Our study advocates two main conclusions. First, NLG1 seems to play an important role in the organization of trans-synaptic 'nano-columns', by positioning AMPAR nanodomains in close proximity to pre-synaptic release sites. Second, the amplitude of AMPAR-mediated currents is highly sensitive to pre-post synaptic nanoscale alignment (*Figure 6F*). At the synapse, NLG1 displays two distinct functions: during synaptogenesis, it binds to pre-synaptic proteins and initiate synapse formation and at

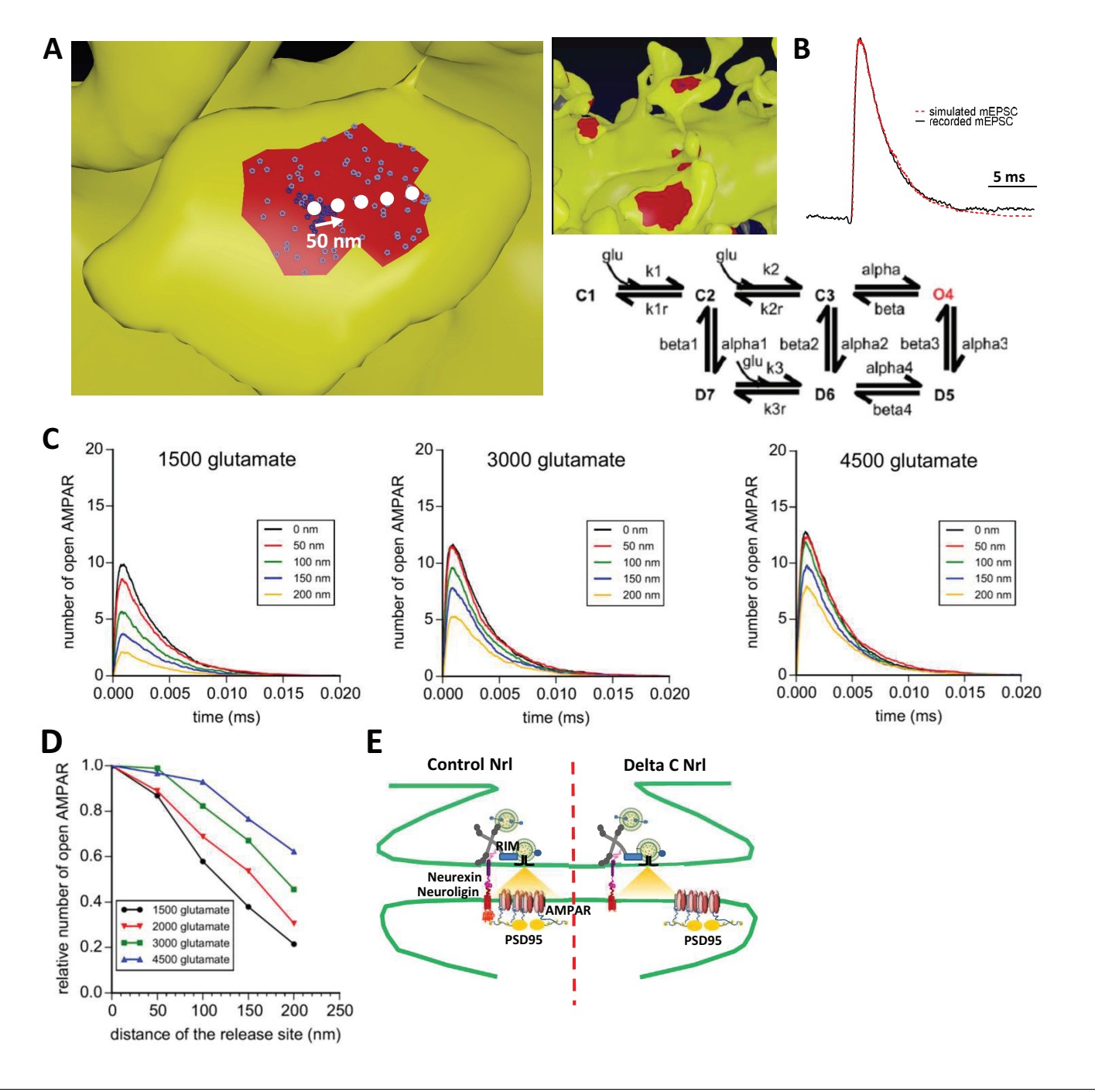

**Figure 6.** Simulation of AMPAR activation. (**A**) View of dendritic spine with synaptic contact area (red patch) containing 25 AMPARs anchored in nanocluster (blue particles) and 70 freely diffusible AMPAR (light blue particles). The simulated glutamate release locations are shown by white dots spaced 50 nm apart. (**B**) Superposition of a miniature average trace obtained by electrophysiology (dark line) and simulated currents (red line) and, kinetic scheme for activation of AMPAR by glutamate (*Jonas et al., 1993*). Kinetic rate constants values (after fitting as described in Materials and methods): $k1 = 13.77\ \mu M^{-1}.s^{-1}$; $k1r = 2130\ s^{-1}$; $k2 = 85.2\ \mu M^{-1}.s^{-1}$; $k2r = 1630\ s^{-1}$; $alpha = 1696\ s^{-1}$; $beta = 630\ s^{-1}$; $alpha1 = 1445\ s^{-1}$; $beta1 = 19.6\ s^{-1}$; $alpha2 = 86\ s^{-1}$; $beta2 = 0.3635\ s^{-1}$; $alpha3 = 8.85\ s^{-1}$; $beta3 = 2\ s^{-1}$; $alpha4 = 6.72\ s^{-1}$; $beta4 = 133.28\ s^{-1}$; $k3 = 3.81\ \mu M^{-1}.s^{-1}$; $k3r = 22.85\ s^{-1}$; AMPAR diffusion constant $= 0.1\ \mu m^2.s^{-1}$; and glutamate diffusion constant $= 100\ \mu m^2.s^{-1}$. (**C**) Time course of simulated AMPAR activation resulting from release of 1500, 3000, and 4500 glutamate molecules at each release location are shown. Each time course is the average of 100 simulations. (**D**) Normalized peak number of open AMPARs activated by release of 1500, 2000, 3000, and 4500 glutamate molecules at each release location is shown. Dashed line at 90 nm indicates data displayed in E. (**E**) Percent decrease in peak number of open AMPAR as a function of number of

*Figure 6 continued on next page*

*Figure 6 continued*

glutamate molecules released, at 90 nm release distance. Dashed lines indicate that when ~2000 glutamate molecules are released the peak number of open AMPARs will be decreased by 25% at a release distance of 90 nm.

The online version of this article includes the following figure supplement(s) for figure 6:

**Figure supplement 1.** Expression of NLG1-ΔCter does not affect paired-pulse ratio.

**Figure supplement 2.** Simulation of AMPAR activation in function of glutamate diffusion properties.

later stages of synapse development NRLG1 organizes the post-synaptic compartment (*Ko et al., 2009*; *Levinson et al., 2005*; *Sara et al., 2005*; *Scheiffele et al., 2000*). NLG1 binds pre-synaptic neurexins via its extracellular domain, while its cytoplasmic tail exhibits various binding sites including a C-terminal PDZ domain binding motif, a central gephyrin binding motif, and an upstream non-canonical motif without identified interactor but also important for neuroligin function (*Dean et al., 2003*; *Giannone et al., 2013*; *Irie et al., 1997*; *Shipman et al., 2011*). Here, we used a NLG1 construct lacking both the PSD-95 and gephyrin binding domains. When expressed during the early stages of synaptogenesis, this NLG1 mutant is able to strongly increase synapse number, but does not recruit enough PSD-95 and AMPARs to sustain normal synaptic transmission (*Budreck et al., 2013*; *Mondin et al., 2011*; *Nam and Chen, 2005*).

## NLG1 is an organizer of the trans-synaptic nano-column

Expression of NLG1-ΔCter later in development (at the synapse maturation stage) caused a modest increase in spine density, accompanied by a decrease in quantal transmission. Interestingly, the AMPAR content per synapse was not affected, nor their diffusion or organization in nanodomains. Rather, NLG1-ΔCter disrupted nanodomain spatial alignment with pre-synaptic release sites visualized by RIM clusters. Our interpretation is that NLG1-ΔCter outcompeted the endogenous NLG1 for the binding to neurexins (or/and other pre-synaptic partners), allowing unanchored PSD-95 scaffolds and connected AMPAR nanodomains to flow away from the release site (*MacGillavry et al., 2013*; *Nair et al., 2013*). As previously published, heterodimerization might occur between recombinant NLG1-ΔCter and endogenous NLG1 (*Shipman et al., 2011*). However, we observed a clear shift between NLG1-ΔCter labelling and AMPAR nanodomains, suggesting a dominant negative effect. Interestingly, a similar reduction in AMPAR currents induced by NLG1-ΔCter was observed in organotypic slices from both NLG1 WT and KO genetic backgrounds, revealing a potential heterodimerization of NLG1-ΔCter with other NLGs, most likely NLG3 (*Shipman et al., 2011*). Acute application of cell-permeant NLG1 C-terminal ligand confirmed these conclusions. Together, these results demonstrate that C-terminal interactions between NLG1 and PDZ domain containing scaffolding proteins such as PSD-95, are important to align AMPAR nanodomains in front of pre-synaptic release sites.

The functional effect of such a molecular disorganization caused either by NLG1-ΔCter expression or NLG1 peptide incubation was found to be unexpectedly important, that is a displacement of AMPAR nanodomain of less than 100 nm decreases mEPSC amplitude by about 30%. Modeling confirmed the high sensitivity of AMPAR-mediated synaptic currents to the position of the pre-synaptic release site with respect to AMPAR nanodomains. Considering that quantal synaptic transmission results from the release of glutamate from one vesicle in front of an AMPAR nanodomain, three parameters determine the number of activated AMPARs: 1/the amount of glutamate per vesicle and its diffusion properties inside the cleft; 2/the number of AMPARs per nanodomain; and 3/the degree of apposition between release sites and nanodomains. Our simulations predict that an equivalent 25% decrease of current could be explained by either a 3-fold decrease in glutamate content (from 4500 to 1500 molecules per vesicle), a 32% loss in the number of AMPARs per nanodomain, or a 100 nm shift between pre- and post-organization (for a glutamate content of 2000 molecules per vesicle), underlying the high sensitivity of synaptic currents to pre-post alignment.

## Pre-post misalignment affects both action potential -dependent and - independent synaptic currents

Interestingly, a similar effect has been observed on miniature EPSCs recorded from primary cultures and both EPSCs and asynchronous EPSCs recorded from organotypic slices. Action potential-dependent currents recorded from slices in calcium or strontium conditions do not directly reflect the

synaptic response because they combine (i) the synaptogenic effect of NLG1 via its extracellular interaction with pre-synaptic adhesion proteins, which affect the parameter n and (ii) its role in AMPAR recruitment and pre-post alignment, affecting the parameter q. Nevertheless, these experiments are necessary to test the potential effect of mis-alignment regarding the pool of vesicle recruited after an action potential, which might differ from the one involved in miniature release (*Fredj and Burrone, 2009*; *Hua et al., 2010*; *Wilhelm et al., 2010*).

The determination of the q from calcium experiments is challenging because EPSC amplitude is dependent on both n, p and q together. First, the absence of effect on the paired pulse properties (*Figure 4—figure supplement 4*) indicates that the p value should not be affected by NLG1 and NLG1-ΔCter expression. Second, the net increase of EPSC amplitude in calcium conditions induced by both NLG1 and NLG1-ΔCter compared to control neurons can be attributed to an increase of the n parameter by recruiting additional pre-synapses, as abundantly described in the literature and shown *Figure 1* (*Figure 4—figure supplement 1*). However, the significant decrease of evoked AMPAR-mediated EPSC amplitude caused by NLG1-ΔCter with respect to NLG1, combined with the increase in CV is more difficult to explain. It can be attributed to either a recruitment of silent synapses by NLG1-ΔCter, or a decrease of the quantal size (q) due for example to pre-post synaptic misalignment.

Strontium experiments represent a tempting way to overcome this multifactorial effect, by decorrelating action potential stimulation and quantal release. Our results show a significant increase of asynchronous EPSC frequency, which can be attributed to the formation/stabilisation of additional synapses, and a decrease of amplitude which likely results from a disruption of the pre-post alignment. However, we cannot exclude the bias due to the origin of the vesicle pool recruited during asynchronous EPSC, which is under debate (*Li et al., 1995*). However, taken together, spontaneous miniature currents recorded in calcium and asynchronous EPSCs recorded in strontium suggest that NLG1-ΔCter expression affects both action potential-dependent and -independent synaptic transmission. Although glutamatergic synapses from cultured systems and brain slices share a similar organization in nanomodules (*Hruska et al., 2018*; *Tang et al., 2016*), it could be interesting to validate our results in more intact preparations.

Our observations, done both on dissociated neurons and organotypic slice cultures, support a new model of synaptic function, where the quantum of synaptic transmission is more sensitive to AMPAR nanoscale organization and alignment with respect to release sites, than to the glutamate content per vesicle and even AMPAR content per nanodomain. These specific properties are due in part to the relatively low affinity of AMPAR for glutamate (mM range) and the fact that released glutamate rapidly fades away laterally (*Lisman et al., 2007*; *Liu et al., 1999*; *Tarusawa et al., 2009*). These results suggest two changes in our conception of the efficacy of synaptic function. First, synapses exhibit a relative tolerance to variability both in the glutamate content per vesicle and in the number of AMPAR per nanodomain. Indeed, a 10–20% variation in these parameters will not drastically affect synaptic transmission efficacy. Our computer simulations predict that the combination of pre-post alignment and nanodomain organization is sufficient to bring some stability to synaptic transmission. Second, fast and large modifications in synaptic amplitude can be better achieved by molecular pre-post misalignment than by changes in AMPAR number. This prediction is in line with the transient physiological misalignment between RIM and PSD95 clusters previously described upon induction of chemical Long Term Depression (LTD) (*Tang et al., 2016*). Interestingly, our modeling data predict that the amplitude of AMPAR currents is not sensitive to the glutamate clearance by glial cells, in agreement with the model that AMPAR activation takes place within the 100 nm surrounding the release site. By varying the gating parameters, we do not observe AMPAR opening beyond 250 nm from the site of vesicle fusion.

Previous models suggested that high efficiency of synaptic transmission, high amplitude response and low variability, requires a tight clustering of AMPA receptors, and an alignment of these clusters with the pre-synaptic release site (*Nair et al., 2013*; *Tarusawa et al., 2009*). Our study proposes that synapses, via NLG1-based trans-synaptic adhesion, optimize the use of glutamate by controlling the alignment between pre-synaptic release sites and AMPAR nanodomains, with surprisingly high sensitivity.

## Materials and methods

### Cell and brain slice culture and transfection

Preparation of cultured neurons was performed as previously described (*Nair et al., 2013*). Hippocampal neurons from 18-day-old rat embryos of either sex were cultured on glass coverslips following the Banker protocol. Neurons were transfected using Effectene (Qiagen) at 10–11 days in vitro (DIV) with either HA::SEP::GluA1, HA::NLG1ΔC, where the last 72 amino acid are truncated, HA:: NLG1 WT or GFP alone and the cells were used for immunocytochemistry at 13–15 DIV. All experiments are done on at least 3 independent dissections.

Organotypic hippocampal slice cultures were prepared from wild type mice (C57Bl6/J strain). Briefly, hippocampi were dissected out from animals at postnatal day 5–7 and slices (350 μm) were cut using a tissue chopper (McIlwain) and incubated with serum-containing medium on Millicell culture inserts (CM, Millipore). The medium was replaced every 2–3 days. After 3–4 days in culture, CA1 pyramidal cells were processed for single-cell electroporation with plasmids encoding enhanced GFP (EGFP) along with wild type or -ΔCter AP-NLG1 (*Chamma et al., 2016*). The pipette containing 33 ng.μl$^{-1}$ total DNA was placed close to the soma of individual CA1 pyramidal neurons. Electroporation was performed by applying three square pulses of negative voltage (10 V, 20 ms duration) at 1 Hz, and then the pipette was gently removed. Three to five neurons were electroporated per slice, and the slice was placed back in the incubator for 7 days before experiments.

### Immunocytochemistry

For the GluA2, HA-tagged proteins and RIM, primary neuronal cultures were co-incubated with monoclonal mouse anti-GluA2 antibody (*Nair et al., 2013*), monoclonal rat anti-HA antibody (Roche, France), RIM 1/2 antibody (synaptic systems, Gottingen, Germany) or for 5–7 min at 37°C and then fixed with 4% PFA. A preliminary permeabilization step with triton is used for the RIM labelling.

Then cells were washed three times for 5 min in 1x PBS. PFA was quenched with NH4Cl 50 mM for 30 min. Unspecific staining was blocked by incubating coverslips in 1% BSA for 1 hr at room temperature. Primary antibodies were revealed with Alexa 532 coupled anti-mouse IgG secondary antibodies and Alexa 647 coupled anti-rat secondary antibodies (Jackson ImmunoResearch Laboratories, USA).

### Direct stochastic optical reconstruction microscopy (d-STORM)

d-STORM imaging was performed on a commercial LEICA SR GSD, model DMI6000B TIRF (Leica, Germany). LEICA SR GSD was equipped with anti-vibrational table used to minimize drift, Leica HCX PL APO 160 × 1.43 NA oil immersion TIRF objective and laser diodes with following wavelength: 405, 488, 532, 642 nm (Coherent, USA). Fluorescence signal was detected with sensitive iXon3 EMCCD camera (Andor, UK). Image acquisition and control of microscope was driven by Leica software. Images were streamed at 94 fps (frames per second); image stack contained typically 30,000 frames. Selected ROI (region of interest) had dimension of 200 × 200 pixels (one pixel = 100 nm). Pixel size of reconstructed super-resolved image was set to 20 nm.

Power of a 405 nm laser controlled the level of single molecules per frame. The dyes were sequentially imaged (Alexa 647 first, followed by Alexa 532) to collect the desired single molecule frames and to avoid photo bleaching. Multi-color fluorescent microspheres (Tetraspeck, Invitrogen) were used as fiducial markers to register long-term acquisitions and correct for lateral drifts and chromatic shifts. A spatial resolution of 14 nm was measured using centroid determination on 100 nm Tetraspeck beads acquired with similar signal to noise ratio than d-STORM single molecule images. Details of experimental procedure and data analysis was followed as described before (*Nair et al., 2013*).

Tesselation analysis of d-STORM experiments are done as described in the original paper (*Levet et al., 2015*).

### uPAINT

13–14 Days in vitro (DIV)-dissociated neurons were imaged at 37°C in an open Ludin Chamber (Ludin Chamber, Life Imaging Services, Switzerland) filled with 1 ml of Tyrode's. Dendritic ROIs were selected based on fluorescence from GFP. ATTO-647 coupled to antibody against AMPAR subunit

GluA2 was added to the chamber after appropriate cell was identified and region selected. Adding a suitable amount of probes controlled density of labelling. The fluorescence signal was collected using a sensitive EMCCD (Evolve, Photometric, USA). Acquisition was driven with MetaMorph software (Molecular Devices, USA) and acquisition time was set to 20 ms. Around 20,000 frames were acquired in typical experiment, collecting up to few thousands of trajectories. Sample was illuminated in oblique illumination mode. Angle of refracted beam varied smoothly and was adjusted manually to maximize signal to noise ratio. The main parameters determined from the experiments were the diffusion coefficient (D) based on the fit of the mean square displacement curve (MSD). Multi-colour fluorescence microspheres were used for image registration and drift compensation. uPAINT data analysis was reported before (*Giannone et al., 2010*; *Nair et al., 2013*).

## Electrophysiology recordings on cell culture

Coverslips of transfected neurons were placed in a Ludin Chamber on an inverted motorized microscope (Nikon Eclipse Ti). Extracellular recording solution was composed of the following (in mM): 110 NaCl, 5.4 KCl, 1.8 $CaCl_2$, 0.8 $MgCl_2$, 10 HEPES, 10 D-Glucose, 0.001 Tetrodotoxin and 0.05 Picrotoxin (pH 7.4; 245 mOsm). Patch pipettes were pulled using a horizontal puller (P-97, Sutter Instrument) from borosilicate capillaries (GB150F-8P, Science Products GmbH) to resistance of 4–6 MΩ and filled with intracellular solution composed of the following (in mM): 100 K-gluconate, 10 HEPES, 1.1 EGTA, 3 ATP, 0.3 GTP, 0.1 $CaCl_2$, 5 $MgCl_2$ (pH 7.2; 230 mOsm). Transfected neurons were identified under epifluorescence from the GFP signal. Recordings were performed using an EPC10 patch clamp amplifier operated with Patchmaster software (HEKA Elektronik). Whole-cell voltage clamp recordings were performed at room temperature and at a holding potential of −70 mV. Unless specified otherwise, all chemicals were purchased from Sigma-Aldrich except for drugs, which were from Tocris Bioscience.

Data were collected and analysis of miniature EPSCs was performed using a software developed by Andrew Penn, the matlab script is available on MATLAB File Exchange, ID: 61567; http://uk.mathworks.com/matlabcentral/fileexchange/61567-peaker-analysis-toolbox).

## Electrophysiology recordings on organotypic brain slices

Dual whole-cell patch-clamp recordings were carried out in the CA1 region from organotypic hippocampal slices placed on the stage of a Nikon Eclipse FN1 upright microscope at room temperature and using Multiclamp 700B amplifier (Axon Instruments). The recording chamber was continuously perfused with aCSF bubbled with 95% $O_2$/5% $CO_2$ and containing (in mM): 125 NaCl, 2.5 KCl, 26 $NaHCO_3$, 1.25 $NaH2PO_4$, 2 $CaCl_2$, 1 $MgCl_2$, and 25 glucose. Resistance of patch-pipettes was 4–6 MΩ when filled with a solution containing (in mM): 135 $CesMeSO_4$, 8 CsCl, 10 HEPES, 0.3 EGTA, 4 MgATP, 0.3 NaGTP, 5 QX-314, pH 7.28, 302 mOsm. EPSCs were elicited in CA1 pyramidal neurons by stimulating Schaffer collaterals in the stratum radiatum with a bipolar stimulation electrode in borosilicate theta glass filled with aCSF. Bicuculline (20 µM) was added to the aCSF to block inhibitory currents and DNQX (100 nM) was added to control epileptic activity. Series resistance was always lower than 20 MΩ. Paired-pulse ratio was determined by delivering two stimuli 50 ms apart and dividing the peak response to the second stimulus by the peak response of the first one. For recordings aEPSCs, extracellular $CaCl_2$ was substituted to equimolar $SrCl_2$. aEPSCs evoked within 500 ms after the stimulation, were analyzed off-line with Mini Analysis software (Synaptosoft). In all cases, at least 20 sweeps per recording were analyzed with a detection threshold set at 5 pA.

## Co-localization analysis

Co-localization analysis was performed using custom written program in Matlab (Mathworks, UK). Manders' coefficients were chosen as co-localization measure because they do not depend on the relative intensity difference between two component images, therefore bypassing alteration in labeling efficiency of different cellular structures. Here, we perform pairwise analysis between coincidental objects observed in two image components. Thus, our Manders' coefficients represent fraction of the intensity belonging to co-localizing super-resolution pixels of a given object. Manders' coefficients are calculated using following equations:

$$M_1 = \frac{\sum_i^n I_{1i,coloc}}{\sum_i^n I_{1i}}$$

$$M_2 = \frac{\sum_i^n I_{2i,coloc}}{\sum_i^n I_{2i}}$$

Where $i$ is a pixel index, 1 and 2 stands for two image components and $n$ is the number of pixels in an object for which coefficient is evaluated. $I_{1i} \geq 0$ ($I_{2i} \geq 0$) are intensity values at the $i^{th}$ pixel of an object in the first (second) component of the dual-color image.

In first step, two image components are threshold, segmented and reduced to sets of geometrical objects attributed with their weighted centroid location, pixel area, intensity and location. Objects in each component image were divided into two categories, according to their area. This distinction is based on the size of single emitter found both on the coverslip and on the dendrite. With this analysis, we can tell apart the single proteins from the clustered ones, and analyze them independently.

In subsequent step, first bivariate nearest neighbor distance distribution is calculated for each neuroligin to the nearest AMPAR cluster. Afterwards, the Manders' coefficients are evaluated between each first nearest neighbor pair of AMPAR cluster and neuroligin. These coefficients were calculated only between pairs of AMPAR and neuroligin separated by the threshold distance, which reflects the maximum distance between two objects considered as related and was obtained from bivariate nearest neighbor distance distribution.

Co-localization significance was accounted for by image randomization. Objects in one image component were rearranged by random assignment of new position for their weighted centroids. This step was repeated up to 1000 times, each time appropriate measure of co-localization was evaluated. Bivariate first nearest neighbour distributions are compared to the mean of randomized samples and 95% confidence intervals. If the experimental distribution lies above (below) randomized distribution, it indicates tendency towards association (dispersion) at given distances. However, if experimental distribution matches randomized one, it points to random or independent distribution between two classes of objects. Matlab scripts are available on request and will be deposited at https://github.com/inatamara/2-Color-dSTORM-object-based-co-localization-with-Manders-coefficients (copy archived at https://github.com/elifesciences-publications/2_color_dSTORM).

## Biomimetic ligands

Synthesis of the divalent TAT-non-sense ligands was previously described in *Sainlos et al. (2011)* (Sainlos et al.). The neuroligin divalent ligands were produced similarly using the last 15 amino acids of NLG1 as PDZ domain binding motifs.

## Modeling

Computer modeling was performed using the MCell/CellBlender simulation environment (http://mcell.org) with MCell version 3.3, CellBlender version 1.1, and Blender version 2.77a (http://blender.org). The realistic model of glutamatergic synaptic transmission (*Figure 6A*) was constructed from 3DEM of hippocampal area CA1 neuropil as described in (*Bartol et al., 2015a*, *Bartol et al., 2015b*; *Kinney et al., 2013*). The 3DEM reconstruction is highly accurate and detailed and contains all plasma membrane bounded components including dendrites, axons, astrocytic glia and the extracellular space itself, in a $6 \times 6 \times 5$ um$^3$ volume of hippocampal area CA1 stratum radiatum from adult rat. As fully described in *Kinney et al. (2013)* special methods were developed and applied to the 3DEM to correct for shrinkage and fixation artefacts to accurately recover the dimensions and topology of the extracellular space. The model contains glutamate transporters, 10,000 per square micron, on the astrocytic glia processes, as described in *Bartol et al., 2015b*. The images in *Figure 6A* were generated from the 3DEM reconstruction using Blender (blender.org) and the CellBlender addon (mcell.org).

The AMPAR rate constants in the model were adjusted using simplex optimization with minimum least-squares to best fit the shape of the AMPAR current (20–80% rise time, peak amplitude, 10–90% fall time of the AMPAR current) reported in *Nair et al. (2013)*. The initial parameter values are as reported in *Jonas et al. (1993)* with the release of glutamate directly over the cluster while holding fixed values of n_Glu = 3000, n_AMPAR = 25 in cluster. The best fit parameter values are reported in the caption for *Figure 6B*. We averaged the AMPAR activation time courses of 100 simulation trials at each release location and number of glutamate released.

## Sampling and statistics

N values represent the number of cells, for each experiment, we used at least four independent dissections.

Summary statistics are presented as mean ± SEM (Standard Error of the Mean). Statistical significance tests were performed using GraphPad Prism software (San Diego, CA). Test for normality was performed with D'Agostino and Pearson omnibus tests. For non-normally distributed data, we applied Mann-Whitney test or Wilcoxon matched-pairs signed rank test for paired observations. When the data followed normal distribution, we used paired or unpaired *t*-test for paired observations unless stated otherwise. ANOVA test was used to compare means of several groups of normally distributed variables. Indications of significance correspond to p values < 0.05(*), p<0.005(**), and p<0.0005(***). After ANOVA analysis, we apply a Dunett's post test to determine the p value between two conditions, results of these tests are noted Anova post-test.

## Ethical approval

All experiments were approved by the Regional Ethical Committee on Animal Experiments of Bordeaux.

## Acknowledgements

We acknowledge E Gouaux for the anti-GluA2 antibody; J-B Sibarita and C Butler for providing single particle analysis software, I Gauthereau for anti-GFP nanobody production; C Breillat, B Tessier, S Benquet and E Verdier for cell culture and plasmid production; E Normand and S Daburon for virus tests; and M Goillandeau and A Penn for mEPSC analysis software (Detection Mini). This work was supported by funding from the Ministère de l'Enseignement Supérieur et de la Recherche (ANR NanoDom and AMPA-T), Centre National de la Recherche Scientifique, FRM to BC, ERC Grants nano-dyn-syn (232942) and ADOS (339541) to DC, the Conseil Régional de Nouvelle Aquitaine and SyMBaD – ITN Marie Curie, Grant Agreement n° 238608 – Seventh Framework Program of the EU to KH.

## Additional information

### Funding

| Funder | Grant reference number | Author |
| --- | --- | --- |
| Agence Nationale de la Recherche | NanoDom | Mathieu Letellier<br>Daniel Choquet<br>Olivier Thoumine<br>Eric Hosy |
| Centre National de la Recherche Scientifique | | Mathieu Letellier<br>Matthieu Sainlos<br>Daniel Choquet<br>Olivier Thoumine<br>Eric Hosy |
| Fondation pour la Recherche Médicale | | Benjamin Compans |
| H2020 European Research Council | nano-dyn-syn | Kalina T Haas<br>Benjamin Compans<br>Dolors Grillo-Bosch<br>Matthieu Sainlos<br>Daniel Choquet<br>Eric Hosy |

The funders had no role in study design, data collection and interpretation, or the decision to submit the work for publication.

### Author contributions

Kalina T Haas, Resources, Software, Formal analysis, Validation, Investigation, Methodology, Writing—review and editing; Benjamin Compans, Formal analysis, Validation, Investigation,

Methodology, Writing—review and editing; Mathieu Letellier, Conceptualization, Validation, Investigation, Methodology, Writing—original draft, Writing—review and editing; Thomas M Bartol, Conceptualization, Resources, Software, Investigation, Writing—review and editing; Dolors Grillo-Bosch, Resources, Methodology; Terrence J Sejnowski, Funding acquisition, Writing—review and editing; Matthieu Sainlos, Conceptualization, Investigation, Writing—review and editing; Daniel Choquet, Conceptualization, Funding acquisition, Writing—review and editing; Olivier Thoumine, Conceptualization, Supervision, Funding acquisition, Writing—original draft, Writing—review and editing; Eric Hosy, Conceptualization, Formal analysis, Supervision, Funding acquisition, Validation, Investigation, Writing—original draft, Project administration, Writing—review and editing

## Author ORCIDs

Benjamin Compans https://orcid.org/0000-0001-7823-1499
Dolors Grillo-Bosch https://orcid.org/0000-0002-8695-5718
Terrence J Sejnowski https://orcid.org/0000-0002-0622-7391
Matthieu Sainlos https://orcid.org/0000-0001-5465-5641
Daniel Choquet http://orcid.org/0000-0003-4726-9763
Eric Hosy https://orcid.org/0000-0002-2479-5915

## Ethics

Animal experimentation: These experiments have been conducted in France, the rat Ratus norvegicus, in accordance with directive 86/609/EEC of 19 October 1986, on the protection of animals used for experimental and other scientific purposes. It was followed in France by 3 rulings in 1987, 2001, 2005 (article R214-87 and R214-130 of the rural code) and 3 rulings of 19 April 1988. An ethical system complemented this regulation since 2008, with harmonization of the commitments between the private and the public sector, by the signature of the National Charter for ethical animal experimentation by institutions in animal experimentation ethics (EAA) and by the CNREEA (Centre national of ethical reflection on animal experiments). Our projects are therefore assessed by the Ethics Committee No 50 of Bordeaux attached to the CNREEA. Thanks to our preparation, we will be ready to apply from January 1St, 2013, the 63-2010-EU directive which transcribed in French law will be published on November 12, 2012 in the form of 2 decree (Code Rural R214-87 to 138) and 4 bylaws.

## Decision letter and Author response

Decision letter https://doi.org/10.7554/eLife.31755.sa1
Author response https://doi.org/10.7554/eLife.31755.sa2

# Additional files

## Supplementary files

• Transparent reporting form

## Data availability

All data generated or analysed during this study are included in the manuscript and supporting files.

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
