## [Decision Letter]

Thank you for submitting your article "Pre-post synaptic alignment through neuroligin tunes synaptic transmission efficiency" for consideration by *eLife*. Your article has been reviewed by two peer reviewers, and the evaluation has been overseen by a Reviewing Editor and Gary Westbrook as the Senior Editor.

The reviewers have opted to remain anonymous. The reviewers have discussed the reviews with one another and the Senior Editor has drafted this decision to help you prepare a revised submission.

As you will see some of the comments will require additional experiments concerning:

1) The validation of the culture results using slice recording;

2) The inclusion of evoked responses in addition to the mEPSC analysis;

3) Additional simulation to address the comments of both reviewers; and

4) Editing to avoid overinterpreting the results.

The original non-overlapping comments of the reviewers are below.

*Reviewer #1:*

The paper by Haas et al. examines the role of neuroligin in aligning postsynaptic receptor nanodomains to presynaptic active zones and the functional consequences of this alignment for synaptic transmission. To address these questions, the authors use imaging methods (dSTORM), electrophysiology, and modeling. The main findings are:

- Super resolution imaging reveals a strong spatial colocalization between AMPA receptors and neuroligin-1;

- Truncation of the C terminus of neuroligin-1 disrupts this colocalization and shifts the release machinery away from AMPA receptor clusters;

- Electrophysiological analysis and modeling suggest that this physical shift markedly decreases the efficiency of synaptic transmission.

Based on these results, the authors conclude that the alignment of pre- and postsynaptic components plays a critical role to ensure the efficiency of synaptic transmission. Overall, I found this a potentially interesting paper. The combination of high-resolution imaging, electrophysiology, and MCell modeling is nice and provides new insights into the mechanisms of synaptic transmission. However, it is also clear that parts of the paper are preliminary, and major revision is required to address these weaknesses.

1) NLG1 and NLG1 deltaC were mostly expressed on wild-type background. This is a complication, because endogenously and exogenously expressed NLG1 mix. The necessary KO control data come only very late in the manuscript, when functional properties of synaptic transmission are assessed.

2) The conclusions stand and fall with the validity of the culture systems used. The authors should make an attempt to validate the results in acute slice preparations.

3) The functional characterization of synaptic transmission is rudimentary. A lot can go wrong in the analysis of miniature synaptic events, and evoked and miniature release may not even be mediated by the same postsynaptic receptors. Thorough analysis of evoked EPSCs seems required to support the conclusions.

4) The diffusion coefficient of glutamate in the synaptic cleft seems an important parameter. However, the value used is unclear (it is just briefly mentioned in the legend of Figure 6), and how its exact value would affect the conclusions remains open. This point should be addressed in systematic simulations. Furthermore, the synaptic cleft height may affect the conclusions. Why was it assumed 20 nm if the authors have the full reconstruction of the extracellular space?

5) The presentation of the results requires improvement. The functional significance of the NLG1-mediated alignment seems sometimes overstated. The relation between receptor alignment and previously reported receptor mobility should be better discussed (otherwise an apparent contradiction may remain for naive readers).

Reviewer #2:

This is a nice work dealing with an important and timely issue, which refers us to the basic mechanisms of synaptic transmission and its efficacy control. The authors employ some cutting-edge super-resolution techniques and electrophysiology in cultured neurons to find that the neuroligin NGL1 plays a key role in the clustering of postsynaptic AMPA receptors in front of the presynaptic glutamate release sites. This finding provides novel and important insights into the molecular machinery underpinning the functioning of brain circuits.

[Editors' note: further revisions were requested prior to acceptance, as described below.]

Thank you for resubmitting your work entitled "Pre-post synaptic alignment through neuroligin tunes synaptic transmission efficiency" to *eLife*. Your revised article has been evaluated by Gary Westbrook (Senior Editor), a Reviewing Editor, and two reviewers.

Both editors and reviewers think that the manuscript has been improved and has the potential to be an important paper. However, we think the functional aspects of the work have not reached acceptability. Specifically the discussion among reviewers and editors concluded that additional data requested in the first reviews should be pursued to fully support the conclusions the authors would like to make. We realize that this reflects additional work for the authors, but in our opinion this work would substantially increase the merits of the work. We hope you will be able to address these concerns in a revised manuscript.

Major points:

1) Regarding further experiments for slice recordings, the authors' contention that fast manipulation is not possible with this system is not compelling. It is indeed possible that viral expression generates expression at a sufficiently fast time scale, before structural changes are initiated. Before the authors have tried, it cannot be claimed that it does not work.

2) Regarding the criticisms concerning the evoked EPSCs, the authors now provide analysis of EPSCs evoked in strontium. However, issues with these data remain. First, strontium is a rather artificial approach to study evoked synaptic transmission. Second, the slow rise time of the evoked EPSCs is 2 ms (Figure 4—figure supplement 2D), which indicates technical problems with these recordings. How the authors can study the functional effects of nanoscale alignment under these recording conditions remains unclear. Finally, the authors try to argue that evoked EPSCs are difficult to interpret, because overlaying effects of receptor alignment and synapse number are difficult to dissect. However, a pessimistic view on the new data shown in the figure for reviewers would be that the central hypothesis is challenged. The authors could perform nonstationary fluctuation analysis of evoked EPSCs to distinguish between changes in N and q. In any case, additional experiments are needed to address the reviewers’ concerns.

3) Regarding the adjustment of rate constants in the AMPAR model (subsection “Synaptic efficiency critically depends on the AMPAR nanodomains to glutamate release sites distance”), it is unclear how microscopic reversibility can be maintained with these changes. Calculating the products of rates for the two cycles in clockwise and anti-clockwise direction in the modified model, they do not match, suggesting that microscopic reversibility is violated. Furthermore, the authors refer to Figure 5, but the relevant numbers are in the legend of Figure 6.

---

## [Author Response]

As you will see some of the comments will require additional experiments concerning:1) The validation of the culture results using slice recording;

As an answer to reviewer 1, our experimental assays require the “acute” disruption of endogenous neuroligin-1 function, by either expressing a NLG1 dominant negative mutant or incubating neurons with competing NLG1 C-terminal peptides, both experiments being performed within a time frame of 48 hrs. Unfortunately, any attempt to obtain short-term expression of the NLG1-ΔC mutant in the brain has failed, because neither in utero electroporation nor virus transfection are compatible with such short-term expression. If mutants were expressed for more than a couple of days in neurons with already present functional synapses or before proper synapses organization, the impairment of neuroligin-1 function would change AMPAR nanodomain organization and affect synaptic function. This renders impossible the specific study of the pre-post alignment effect because it will likely be contaminated by other synaptic modifications.

The incubation of acute slices with NLG1 TAT-peptides would be an alternative, but the slices are likely to die in a few hours before the pre-post synaptic misalignment takes place. Injecting the peptides in vivo could also be considered, but this would be extremely demanding in the amount of peptides needed (and we have a limited supply). So at this stage, we believe that we cannot provide an alternative to culture systems to demonstrate the effects of NLG1 perturbation on pre-post synaptic alignment.

2) The inclusion of evoked responses in addition to the mEPSC analysis;

Evoked responses differ from at least two aspects compared to spontaneous mEPSCs:

i) They potentially do not recruit the same pool of glutamate vesicles at the pre-synapse (this fact is still highly debated, see Fredj and Burrone, 2009, Hua et al., 2010; Wilhelm et al., 2010, Tang et al., 2016). If this is true, it is possible that release sites responsible for evoked and miniature currents are different from each other. We understand the concern of the reviewers regarding the role of Nlg1 and pre-post alignment in evoked transmission. However, miniature responses recorded in the presence of strontium, presented in the paper, correspond to “evoked responses” in the sense that they are triggered by action potentials and likely involve the vesicle pool dedicated to calcium-induced release. Importantly, we observed a similar effect of Nlg1-ΔC expression on miniature AMPA currents recorded in primary neurons and on evoked asynchronous miniature EPSCs in the presence of strontium, suggesting that the Nlg1-ΔC misaligns both types of release sites (if they are different) from AMPAR nanodomains. Computer simulations showing the influence of a misalignment between glutamate release sites and AMPAR nanodomains further validate our experimental approaches. This point is now discussed in the Discussion section.

ii) Evoked responses strongly depend on the number of active synapses (n), a parameter that is known to be altered when manipulating Nlg1 expression or function (e.g., Chubykin et al., Neuron, 2007; Mondin et al., 2011; Giannone et al., 2013; Chih et al., 2005). In particular, when expressed for several days, Nlg1-deltaC is known to increase the number of pre-synapses formed on transfected neurons (e.g., Mondin et al., 2011), in addition to the effect we see on the pre-post alignment. This is the reason why we found difficult to extract with precision the quantal size from EPSCs amplitudes and opted for the analysis of spontaneous and evoked asynchronous miniature events. By analysing the synchronized fraction of evoked currents recorded in strontium (organotypic slices from WT or KO mice), we found that the synaptogenic activity of Nlg1-deltaC depends on the genetic background (WT or KO), which further complicates the analysis and interpretation of results (see response to reviewer 1 for more details and also Shipman et al., 2011). In addition, we performed a new set of experiments to study evoked currents in cell cultures. However, neither the amplitude nor the kinetics of EPSCs show differences between NLG1 WT and the delta C mutant. Those results are included in the response to reviewer 1.

For all these reasons we would rather not complicate the message of the paper by entering into the highly documented role of neuroligin in the synaptogenesis, and would like to maintain the focus of the paper on the pre-post misalignment at single synapses.

3) Additional simulation to address the comments of both reviewers; and

All requested simulations have been done, addressing the impact of glutamate diffusion rate (requested by reviewer 1) and the role of environment (requested by reviewer 2). These simulations have been added in the manuscript as supplemental figures and are discussed in the text.

4) Editing to avoid overinterpreting the results.

We took into account the comment of the reviewer 1 and tried to soften some of our conclusions.

We thank the editor to give us the opportunity to improve the manuscript and we hope that this new revised manuscript will convince both the editor and the reviewers.

The original non-overlapping comments of the reviewers are below.Reviewer #1:[…] 1) NLG1 and NLG1 deltaC were mostly expressed on wild-type background. This is a complication, because endogenously and exogenously expressed NLG1 mix. The necessary KO control data come only very late in the manuscript, when functional properties of synaptic transmission are assessed.

We thank the reviewer for raising this important point. Although we understand that the KO control might appear as necessary, we want to emphasize that the whole strategy of our study was to challenge the role of Nlg1 in already formed synapses by using a dominant-negative strategy. We achieved this goal by using either a Nlg1 truncation mutant or a competing divalent peptide that we introduced in neurons as acutely as possible to limit any effect related to the synaptogenic activity of Nlg1.

However, in addition to the electrophysiology experiments carried out in organotypic slices, we initially performed super-resolution imaging experiments in cultured neurons from Nlg1 KO mice (not shown in the paper) and found that the number of AMPARs per nanodomain was significantly decreased compared to wild-type neurons (see Author response image 1). Interestingly, these results are in agreement with the recent report that AMPARs at the plasma membrane are decreased in cultured hippocampal neurons lacking Nlg1/2/3 (Chanda et al., J Neurosci, 2017). Furthermore, both in vitro and in vivo studies have shown that NMDAR-mediated transmission is decreased in Nlg1 KO neurons and that NMDAR-dependent LTP is impaired (e.g., Varoqueaux et al., Neuron, 2006; Jiang et al., Mol Psy 2016; Chanda et al., J Neurosci, 2017). Overall, these data suggest that synapses in KO neurons do not maturate as well as in the wild-type condition and this is the reason why we think that dominant-negative approaches were more appropriate for addressing our question.

Author response image 1 shows an analysis of the number of AMPA receptors per nanodomain in a WT or NLG1 KO background, showing that the knock-out of NLG1 decreases the composition of AMPAR nanodomain. Incidentally, we previously reported defects in AMPAR recruitment in CA1 neurons of acute NLG1 KO slices (p6-p8) by immunohistochemistry, and a decrease in both the amplitude and frequency of AMPAR-mediated mEPSCs (Mondin et al., 2011), that might be related to this phenomenon.

Interestingly, the fact that we found similar effect of the Nlg1-deltaC expression on evoked quantal events recorded from KO or WT organotypic slices suggest that some other adhesion molecules (most likely Nlg3) compensate to some degree for the absence of Nlg1 in KO neurons (see also Chanda et al., J Neurosci, 2017). In those neurons, Nlg1-deltaC might compete (and possibly heterodimerize) with Nlg3 for binding to PSD-95 at glutamatergic synapses.

2) The conclusions stand and fall with the validity of the culture systems used. The authors should make an attempt to validate the results in acute slice preparations.

We did make some attempts to validate our results in acute slices but we were confronted to two main difficulties:

i) For imaging experiments, the pointing accuracy in slices is deteriorated. We do not consider possible at this point to obtain two colour super-resolution images able to reveal a pre-post misalignment in the order of 100 nm in thick tissue. In our group, we are actively working to towards this goal but it will likely take another few years to reach it.

ii) For electrophysiological experiments, as detailed in the previous answer, the difficulty of this paper consists to express neuroligin as acutely as possible to avoid the bias due to misformation/ maturation of synapses depending on neuroligin. Our conclusions are based on the fact that we are able to demonstrate that AMPAR nanodomains are not modified when delta C is expressed for a couple of days. Technically, we do not know how to express acutely a construct in acute brain slices. The classical methods, either post-natal viral infection (after synaptogenesis) or in utero electroporation (before synaptogenesis) necessitate more than 10 days of expression and are therefore not appropriate. The use of TAT-peptides is unfortunately not an option either as the injection of peptides in animals would require amounts of peptides that we are not able to produce.

Unfortunately, for all these reasons, we do not consider as easily feasible the transfer of our conclusions to an acute system.

3) The functional characterization of synaptic transmission is rudimentary. A lot can go wrong in the analysis of miniature synaptic events, and evoked and miniature release may not even be mediated by the same postsynaptic receptors. Thorough analysis of evoked EPSCs seems required to support the conclusions.

We appreciate this important remark. However, the difficulty with studying evoked-EPSCs rather than miniature events amplitude is that the former strongly depends on the number of stimulated synapses (n), a parameter which is greatly affected when Nlg1 expression is manipulated, as shown figure 1 and reported in the literature (e.g., Chubykin et al., Neuron, 2007; Mondin et al., 2011; Giannone et al., 2013; Chih et al., 2005). As a result, we found difficult to isolate the effect of Nlg1 on the quantal size parameter (q) from evoked-EPSCs and we therefore opted for analysing spontaneous or evoked asynchronous miniature currents instead.

To try to take into account the potential differences between spontaneous and evoked currents, which might engage separate presynaptic pools of vesicles (this fact is still highly debated, see Fredj and Burrone, Nat neurosci, 2009, Hua et al., 2010; Wilhelm et al., 2010, Tang et al., 2016), we decided to analyse both spontaneous miniature currents in primary cultures (in Ca^2+^) and evoked miniature currents in organotypic slices (in Sr^2+^). Both experiments gave similar results, showing that q is decreased when overexpressing Nlg1-deltaC. This suggests that whether spontaneous and evoked transmission engage different pools of vesicles or not, they are nonetheless similarly affected by the Nlg1-deltaC mutant. Furthermore, the fact that both peptide incubation and simulation also give similar results makes us confident that our experimental approach is valid. This point relative to the type of miniature or evoked responses is now discussed in the main text.

In parallel, we performed new analyses and experiments:

i) In organotypic slices from WT or KO mice, we analysed the amplitude of the first peak of the responses evoked in Sr^2+^, which corresponds to the summation of synchronized events. We found that overexpressing Nlg1-WT increases the amplitude of this peak on both the WT or KO background, an effect that we assume to be related to an increase in synapse number (see for instance Shipman et al., 2011; Bemben et al., Nat Neurosci, 2014; Levy et al., 2015). Overexpressing the Nlg1-deltaC mutant had a similar effect on the WT background but seemed less efficient to promote this effect on the KO background (although not statistically significant). This dependence on the presence of endogenous Nlg1 has been described previously (Shipman et al., 2011) and highlights the difficulty to extract the effect of Nlg1 on the quantal size when analysing EPSCs, which depends also on n. This point is interesting but it seems out of the paper focus, which concerns the role of neuroligin in the molecular pre-post organization. For these reasons we would prefer to not to show these data in the article as they may complicate the message.

**Author response image 2. respfig2:** EPSC amplitudes on brain slices (evoked in strontium).

ii) In cell cultures, we recorded spontaneous, action potential-dependent, currents in the absence of TTX after 2 days of Nlg1-WT or mutant expression. We analysed both the amplitude and the kinetics of EPSCs (larger than 50 pA to exclude miniature events) but found no difference across conditions (see Figure below). However, the amplitude data are difficult to interpret because they highly depend on the number of synapses formed by a single axon on the recorded neuron. This parameter is highly variable because it depends for example on the density of cells in the culture and their axonal and dendritic growth rate, etc.

**Author response image 3. respfig3:** Properties of EPSCs in cell culture.

4) The diffusion coefficient of glutamate in the synaptic cleft seems an important parameter. However, the value used is unclear (it is just briefly mentioned in the legend of Figure 6), and how its exact value would affect the conclusions remains open. This point should be addressed in systematic simulations. Furthermore, the synaptic cleft height may affect the conclusions. Why was it assumed 20 nm if the authors have the full reconstruction of the extracellular space?

As requested by the reviewer, we performed modelling experiments with glutamate diffusion coefficients of 300 and 500 µm²/s. Results are reported in the Author response image 4. With these values, the effect of pre-post misalignment is stronger than with a diffusion coefficient of 100 µm²/s, and the number of open AMPAR in response to glutamate release is small, particularly for a limited amount of glutamate per vesicle. We know that glutamate diffusion can be slowed down in a dense environment. Based on our simulation, the use of a diffusion coefficient of 100 µm²/s inside the cleft gave us results more conform to the measured one with electrophysiology. These new modelling results are now added as a supplemental figure to the manuscript.

**Author response image 4. respfig4:** 

Concerning the synaptic cleft, the structure comes from the digitization of the neuropil taken from electron microscopy images. Briefly, as described in the 2 Bartol papers from 2015 cited in the text, the group of Kristen Harris succeeded to reconstruct a complete piece of hippocampus by connecting successive electron microscopy sections. All these results have been subsequently digitized to deliver a complete 3D image of this piece of brain. Even if not represented on our figure, the model takes into account the exact localization of the pre-synapse, of the surrounding environment as glial cells, since all this information are part of the electron microscopy images.

Moreover, on the image, the PSD can be distinguished: it corresponds to the red area inside the synapses. The size of the cleft is directly obtained from electron microscopy, and even if fixation can create small artefacts, we believe that it won’t affect too drastically the model and the simulated results.

5) The presentation of the results requires improvement. The functional significance of the NLG1-mediated alignment seems sometimes overstated.

Some sentences have been changed, see details below:

Previously: These results demonstrate the tight nanoscale co-organization between AMPAR and NLG1 within a synapse.

Now: These results reveal a fairly tight nanoscale co-organization between AMPAR and NLG1 within the synapse.

Previously: Only 38% of NLG1-ΔCter co-localized at least partially with AMPARs, compared with 80% for NLG1, revealing that the NLG1 C-terminal truncation strongly decreases the association between NLG1 and AMPAR nanodomains.

Now: Only 38% of NLG1-ΔCter co-localized at least partially with AMPARs, compared with 80% for NLG1, revealing that the NLG1 C-terminal truncation significantly decreases the association between NLG1 and AMPAR nanodomains.

Previously: Control non-sense ligands had the same structure but mutations in the sequence prevent interaction with PDZ domains. Incubation of 14 DIV hippocampal neurons for 1-2 days with NLG1 competing ligands competing ligands caused a misalignment between RIM and GluA2 containing AMPARs observed by dual-color STORM

Now: Control non-sense ligands had the same structure but mutations in the sequence prevent interaction with PDZ domains. Incubation of 14 DIV hippocampal neurons for 1-2 days with NLG1 competing ligands decreased the co-localization between RIM and GluA2 containing AMPARs observed by dual-color STORM

Previously: First, NLG1 is one of the main organizers of trans-synaptic “nano-columns”

Now: First, NLG1 seems to play an important role in the organization of trans-synaptic “nano-columns”

The relation between receptor alignment and previously reported receptor mobility should be better discussed (otherwise an apparent contradiction may remain for naive readers).

We added a sentence in the Results to explain it: “We previously showed that AMPARs exhibit two diffusion profiles: the immobile fraction represent mostly AMPARs trapped by PSD scaffolding proteins in nanodomains, whereas the mobile fraction represents individual freely moving AMPARs (Nair et al., 2013). This lateral mobility is dependent on AMPAR complex composition, phosphorylation status and desensitization properties (Compans et al., 2016; Constals et al., 2015; Hafner et al., 2015; Tomita et al., 2007).”.

[Editors' note: further revisions were requested prior to acceptance, as described below.]

Both editors and reviewers think that the manuscript has been improved and has the potential to be an important paper. However, we think the functional aspects of the work have not reached acceptability. Specifically the discussion among reviewers and editors concluded that additional data requested in the first reviews should be pursued to fully support the conclusions the authors would like to make. We realize that this reflects additional work for the authors, but in our opinion this work would substantially increase the merits of the work. We hope you will be able to address these concerns in a revised manuscript.Major points:1) Regarding further experiments for slice recordings, the authors' contention that fast manipulation is not possible with this system is not compelling. It is indeed possible that viral expression generates expression at a sufficiently fast time scale, before structural changes are initiated. Before the authors have tried, it cannot be claimed that it does not work.

To answer this comment, we undertook a lentivirus-based strategy to express either HA-tagged NLG1 or NLG-ΔCter in young rats, and then perform acute slice recordings as suggested by the reviewer. The two DNA constructs that were made for this purpose are under a synapsin promoter to specifically target neuronal cells, and comprise an IRES sequence downstream of NLG1, allowing the expression of a GFP reporter. The two constructs were tested by transfection on dissociated cell cultures, before viral encapsulation: high GFP signals were observed that correlated with anti-HA labelling in the same neurons (see below), revealing the expression of the neuroligin-1 protein. We then inserted the plasmids in lentiviral vectors and carried out the production of lentivirus: the titration was 10^9^ viral particles per mL. However, we obtained only sparse and very weak GFP signal on cell cultures with both NRL1 and deltaC viruses. Then, 8 rats (4 rats for each construct) were infected by hippocampal in vivo injection. Unfortunately, acute hippocampal slices 19 and 24 days after injection did not present any GFP signal. Slices were then fixed and immuno labelled with anti-HA antibody, which did not give any signal either.

**Author response image 5. respfig5:** Example of neurons labelled with anti-HA alexa647nm in culture transfected by either Nlg1HA-IRES-GFP or NLG1HA-ΔCter IRES GFP plasmids, used for lentivirus production.

The total size of the plasmid is 11 000 base pairs, which can explain the difficulty of encapsulation into viral particles. Thus, we fear that the main part of produced particles do not contain the inserted plasmid. In the recent literature, the only examples of virus infection used to modulate neuroligin expression are based on the infection of conditional neuroligin floxed mice with lentiviruses expressing cre-recombinase, a at least 2kb smaller insert (Chanda et al. Journal of Neuroscience 2017; Jiang et al., Molecular Psychiatry 2016).

We now clearly specify in the text that experiments were done on cultured systems, and added one sentence to discuss the difference between acute and cultured slices. The implementation of another strategy to try to replicate our results on an acute system would take several additional months without guarantee of success. We hope that the reviewer will acknowledge the time and efforts we spent to test in vivo expression of neuroligin-1 using viruses, and understand the limitations of that strategy for this particular construct.

2) Regarding the criticisms concerning the evoked EPSCs, the authors now provide analysis of EPSCs evoked in strontium. However, issues with these data remain. First, strontium is a rather artificial approach to study evoked synaptic transmission.

We modified the text to specify that the recruited pool of pre-synaptic vesicles in the presence of strontium is still under debate (subsection “Pre-post misalignment affects both action potential -dependent and -independent 382 synaptic currents”, third paragraph). Moreover, as suggested by the reviewer, we have performed more dual patch clamp experiments in organotypic slices cultures, now recording evoked AMPAR-mediated EPSCs in CA1 neurons in the presence of calcium, a more physiological condition than strontium. These new data (appearing in Figure 4—figure supplement 2) show that Nlg1ΔCter expression does not produce such a strong increase in AMPAR-mediated EPSCs as Nlg1 WT, as compared to control non-electroporated neurons. These data thus support our measurements of miniature EPSCs and evoked EPSCs in the presence of strontium. Paragraphs mentioning those data and discussing evoked EPSCs in strontium and calcium conditions have been added in the Results and Discussion, respectively. The way to interpret these data is further discussed below (subsection “NLG1 C-terminus truncation impairs synaptic transmission”; and subsection “Pre-post misalignment affects both action potential -dependent and -independent synaptic currents”).

Second, the slow rise time of the evoked EPSCs is 2 ms (Figure 4—figure supplement 2D), which indicates technical problems with these recordings. How the authors can study the functional effects of nanoscale alignment under these recording conditions remains unclear.

We agree with the reviewer that the rise-time that we measured, which is about 2 ms is relatively slow compared to what have been reported in the literature for other central synapses including the CA3-CA1 synapse. However, several aspects have to be taken into consideration, which we believe relate more to the analysis method and the model than the quality of the recordings per se:

- We reanalysed our traces using the miniAnalysis software and measured the 20-80% rise time, which is more often used in the literature as compared to the 10-90% rise time (that we showed in the first version of the manuscript). Indeed, the 20-80% rise-time is less subject to fluctuations due to the noise that can be found at the onset and peak of events, especially when amplitudes are small. This new analysis gives average values which distribute around 1.5 ms and correspond to the normal high range of what has been published in the literature (see for instance Grillo et al., Neuron, 2018 for CA1 neurons) (see table below).

10-90% rise time (ms)20-80% rise time (ms)Control2,31 ± 0,171,59 ± 0,14NLG1-WT2,09 ± 0,161,36 ± 0,17NLG1-ΔC2,4 ± 0,161,52 ± 0,17

- We plotted the distribution of 10-90% rise-times for individual events recorded in the same cell (see Author response image 5). As you can see in Author response image 5, we could detect at least 3 populations of events: those displaying relatively fast (~1.3 ms), intermediate (~3 ms) and slow (~4.5 ms) rise-times. Note that fast events (<0.5 ms) could be detected in our recordings. To what populations of synapses correspond these events is not clear but one should consider that rise-times depend not only on the functional organization of synapses and the biophysics of receptors but also on the location of the stimulated synapses on the dendritic arbor. As a matter of fact, this parameter is often used to estimate the distance of active synapses to the soma as responses generated by distal synapses will be more filtered compared to responses from proximal synapses (see for instance Smith et al., J Physiol, 2003; Grillo et al., Neuron 2018; Balu et al., J Neuroscience 2007). In organotypic slices, it is very likely that the spatial arrangement between stimulated CA3 axons and CA1 neurons and the topography of connections is modified compared to acute slices, and might explain this heterogeneity.

**Author response image 6. respfig6:** 

Finally, we want to emphasize that strontium experiments have been performed in a paired patch-clamp paradigm, meaning that effect of Nlg1ΔCter expression on asynchronous currents is directly comparable to the control EPSCs recorded from an adjacent non-electroporated neuron, reinforcing the idea that modification of their amplitude is indeed due to the expression of Nlg1 constructs and not to patch-clamp artefacts.

Finally, the authors try to argue that evoked EPSCs are difficult to interpret, because overlaying effects of receptor alignment and synapse number are difficult to dissect. However, a pessimistic view on the new data shown in the figure for reviewers would be that the central hypothesis is challenged. The authors could perform nonstationary fluctuation analysis of evoked EPSCs to distinguish between changes in N and q. In any case, additional experiments are needed to address the reviewers’ concerns.

As requested by the reviewer, we performed additional experiments and recorded EPSCs in Ca²^+^-containing aCSF from CA1 neurons in organotypic slices. Taking advantage of our dual recordings approach, we decided to compare EPSCs from a neuron expressing recombinant NLG1 (WT or ΔCter) and a nearby non-electroporated neuron upon stimulating the same afferent fibers. We found that overexpressing NLG1-WT or NLG1-ΔCter both increase AMPAR-EPSCs amplitude relatively to the control neuron, consistent with the established capacity of NLG1 to increase the number of synaptic contacts with afferent fibers (Mondin et al., 2011; Chih et al., 2005). However, we found that the normalized EPSC amplitude measured from neurons expressing the NLG1-ΔCter mutant was significantly lower compared to neurons expressing NLG1, which in principle can be explained by a smaller number of functional synapses (n), release probability (p) or quantal size (q). From our previous study (Mondin et al., 2011), expression of NLG1-ΔCter recruits presynaptic terminals to the same extent as NLG1, most likely through the binding to presynaptic Neurexins. This is not surprising given that both constructs share the same extracellular domain. Moreover, our recordings of the paired-pulse ratio do not exhibit any difference between control neurons and neurons expressing NLG1-WT or NLG1-ΔCter, indicating no difference at the pre-synaptic level. Therefore, we propose that the difference of EPSC amplitudes between NLG1-WT and NLG1-ΔCter conditions results from a lack of AMPAR recruitment in front of glutamatergic release sites, most likely due to the inability of NLG1-ΔCter to interact with PDZ domain containing including PSD-95 and S-SCAM. Such a lack of recruitment for NLG1-ΔCter compared to NLG1-WT could lead to either the complete silencing of synapses (decreased n) and/or to the misalignment of presynaptic release sites with respect to AMPAR nanodomains (decreased q).

To test these ideas further and discriminate between these two possibilities, we considered performing different types of quantal analyses, including recording responses to minimal stimulations, performing variance-mean analysis (Clements, J. Neurosci Methods, 2003; Silver, J Neurosci Methods 2003) and analysing the coefficient of variation (CV) over many trials taking advantage of our dual recordings configuration (Levy et al., 2015). Minimal stimulations has been tested but they presented strong disadvantages as the restricted sampling to a subset of synapses, the recurrent asynchronous poly-synaptic responses and the delayed responses (see Author response image 7), we decided to not go further with this option. Moreover, we found that the variance-mean analysis which allows to access n and q by varying p with different levels of extracellular calcium (Silver, 2003; Clements, 2003) was highly challenging to apply in organotypic slices due to the high excitability of the circuit. Typically, raising extracellular calcium to 5 mM caused bursts of activity, thus making impossible the analysis of EPSC amplitude and variance (see traces in Author response image 7). We therefore opted for analysing the CV which is possible thanks to our dual recording approach where the electroporated neuron is normalized to a control neuron (Levy et al., 2015).

**Author response image 7. respfig7:** 

The CV is commonly considered as a parameter depending on quantal content (n*p) but not quantal size according to the quantal theory. For instance, comparing the CV in dual recordings enabled Levy et al. (2015) to show the relative loss of functional glutamatergic synapses upon MAGUK knock-down in organotypic slices, consistent with a decrease in the number of asynchronous EPSCs elicited in strontium. However, besides the established contribution of quantal content to CV, several lines of evidence suggest that the dynamic nanoscale organization of AMPARs can also contribute to the CV at the level of individual synapses. For instance, the CV of EPSCs recorded from pairs of neurons is decreased upon crosslinking surface AMPARs with antibodies (Heine et al., 2008), suggesting that AMPAR fluctuations at the synapse due to diffusional exchange can also affect the CV. Furthermore, simulations predict that the CV is inversely correlated with the number of synaptic AMPARs (Czondor et al., 2012). Therefore, while an increase in the parameter n is expected to correlate with a lower CV, one could predict that a lack of PSD-95 dependent anchorage at NRX-NLG adhesion sites produces a higher CV.

We applied the CV analysis in our dual recordings and found that NLG1-WT expression consistently decreases the CV relatively to control neurons, in parallel of increasing the EPSC amplitude. This could be interpreted as an increase in the number of functional synapses, which is in agreement with the increase of asynchronous events elicited in strontium. However, we did not find such a decrease in CV for neurons expressing NLG1-ΔCter, despite the 200% increase in EPSC amplitude. The higher CV found for neurons expressing NLG1-ΔCter compared to NLG1 could be explained either by a higher proportion of silent synapses (n) and/or a higher number of functional synapses in which the pre/post synaptic misalignment induces higher fluctuations. Considering our strontium recordings showing that NLG1-ΔCter expression increases the number of asynchronous events compared to the control neuron but with smaller amplitudes, we believe that the second option is highly plausible. The calcium experiments have been now added as Figure 4—figure supplement 2 and results are both explained in the main Results and discussed in the Discussion (subsection “NLG1 C-terminus truncation impairs synaptic transmission”; and subsection “Pre-post misalignment affects both action potential -dependent and -independent synaptic currents”).

3) Regarding the adjustment of rate constants in the AMPAR model (subsection “Synaptic efficiency critically depends on the AMPAR nanodomains to glutamate release sites distance”), it is unclear how microscopic reversibility can be maintained with these changes. Calculating the products of rates for the two cycles in clockwise and anti-clockwise direction in the modified model, they do not match, suggesting that microscopic reversibility is violated. Furthermore, the authors refer to Figure 5, but the relevant numbers are in the legend of Figure 6.

The reviewer is right, there was a clear mistake in our model parameters. We corrected the values with respect to microscopic reversibility. New parameters have been written in the text. We have performed again all simulations with the corrected values and changed the figures accordingly. Moreover we added a graph where we superimposed a mean recorded trace of miniatures and the mean simulated trace to verify that new parameters fit with physiological data.

Here are the new values with notation based on the model by Jonas et al. 1993: k1 = 13.77 μs^-1^M^-1^s^-1^; k1r = 2130 s^-1^; k2 = 85.2 μs^-1^M^-1^s^-1^; k2r = 1630 s^-1^; alpha = 1696 s^-1^; beta = 630 s^-1^; alpha1 = 1445 s^-1^; beta1 = 19.6 s^-1^; alpha2 = 86 s^-1^; beta2 = 0.3635 s^-1^; alpha3 = 8.85 s^-1^; beta3 = 2 s^-1^; alpha4 = 6.72 s^-1^; beta4 = 133.28s^-1^; k3 = 3.81 μs^-1^M^-1^s^-1^; k3r = 22.85 s^-1^.